# EHD1-dependent traffic of IGF-1 receptor to the cell surface is essential for Ewing sarcoma tumorigenesis and metastasis

Sukanya Chakraborty[1,2], Aaqib M. Bhat[1,2], Insha Mushtaq[1,9], Haitao Luan[1], Achyuth Kalluchi[2], Sameer Mirza[2,10], Matthew D. Storck[1], Nagendra Chaturvedi [3], Jose Antonio Lopez-Guerrero [4], Antonio Llombart-Bosch[5], Isidro Machado[5], Katia Scotlandi [6], Jane L. Meza[7,8], Gargi Ghosal[2,8], Donald W. Coulter[3,8], M. Jordan Rowley [2,8], Vimla Band[2,8], Bhopal C. Mohapatra [2,8✉] & Hamid Band [1,2,7,8✉]

Overexpression of the EPS15 Homology Domain containing 1 (EHD1) protein has been linked to tumorigenesis but whether its core function as a regulator of intracellular traffic of cell surface receptors plays a role in oncogenesis remains unknown. We establish that EHD1 is overexpressed in Ewing sarcoma (EWS), with high EHD1 mRNA expression specifying shorter patient survival. ShRNA-knockdown and CRISPR-knockout with mouse *Ehd1* rescue established a requirement of EHD1 for tumorigenesis and metastasis. RTK antibody arrays identified IGF-1R as a target of EHD1 regulation in EWS. Mechanistically, we demonstrate a requirement of EHD1 for endocytic recycling and Golgi to plasma membrane traffic of IGF-1R to maintain its surface expression and downstream signaling. Conversely, EHD1 overexpression-dependent exaggerated oncogenic traits require IGF-1R expression and kinase activity. Our findings define the RTK traffic regulation as a proximal mechanism of EHD1 overexpression-dependent oncogenesis that impinges on IGF-1R in EWS, supporting the potential of IGF-1R and EHD1 co-targeting.

[1] Eppley Institute for Research in Cancer and Allied Diseases, University of Nebraska Medical Center, Omaha, NE 68198, USA. [2] Department of Genetics, Cell Biology & Anatomy, College of Medicine, University of Nebraska Medical Center, Omaha, NE 68198, USA. [3] Department of Pediatrics, University of Nebraska Medical Center, Omaha, NE 68198, USA. [4] Laboratory of Molecular Biology, Fundación Instituto Valenciano de Oncología, Valencia, Spain. [5] Department of Pathology, University of Valencia, Avd. Blasco Ibáñez 15, 46010 Valencia, Spain. [6] Laboratory of Experimental Oncology, IRCCS Istituto Ortopedico Rizzoli, Bologna, Italy. [7] Department of Biostatistics, College of Public Health, University of Nebraska Medical Center, Omaha, NE 68198, USA. [8] Fred & Pamela Buffett Cancer Center, University of Nebraska Medical Center, Omaha, NE 68198, USA. [9] Present address: Incyte Corporation, Wilmington, DE, USA. [10]Present address: Department of Chemistry, College of Science, United Arab Emirates University, Al Ain, UAE. ✉email: bmohapat@unmc.edu; hband@unmc.edu

Members of the EPS15 homology domain-containing (EHD) protein family (EHD1-4) of membrane-activated ATPases have emerged as key regulators of vesicular traffic along the endocytic pathway[1–3]. Among them, EHD1 has been investigated the most and is well-established to regulate the post-endocytic recycling of a variety of cell surface receptors back to the cell surface[1–3]. In contrast to this role in post-endocytic receptor traffic, our recent studies identified a unique role for EHD1 in the pre-activation transport of newly-synthesized RTKs, CSF1 receptor[4] and EGFR[5] from the Golgi to the plasma membrane to enable their efficient ligand-induced signaling and biological responses. These cell biological findings raise the possibility that overexpression of EHD1 in tumors could promote RTK-dependent oncogenic signaling by enabling the cell surface display of RTKs on tumor cells. This idea is consistent with recent findings of EHD1 overexpression in various cancers, often correlating with shorter survival, and the cell-based studies using gene knockdown or overexpression strategies that support the role of EHD1 overexpression to promote tumorigenesis, chemotherapy resistance, epithelial-mesenchymal transition, stem cell behavior, and glycolysis in various tumor models[6–15]. These studies have linked EHD1 overexpression to distal signaling alterations such as the activation of NFkB, β-catenin, and c-Myc pathways that are not immediately linked to EHD1's core vesicular traffic roles in endocytic recycling and Golgi to cell surface RTK traffic. Consistent with the potential of EHD1 expression in fact regulating the RTK traffic in tumors, EHD1 levels in non-small cell lung cancer correlated with EGFR expression and specified shorter survival, metastasis, and chemotherapy resistance[8,16]. EHD1 was also shown to promote erlotinib resistance in EGFR-mutant lung cancers[11]. However, direct evidence for regulation of RTK traffic as a proximal mechanism to activate the various distal signaling axes in EHD1-overexpressing cancers is currently lacking. Such a linkage is of considerable interest since receptor tyrosine kinases (RTKs) are well-established as oncogenic drivers or as key secondary components of oncogenic programs of other driver oncogenes across cancers[17].

The oncogenesis-associated overactivity of RTKs has been ascribed to multiple mechanisms, including gene amplification, increased transcription, genetic aberrations such as chromosomal translocation, point mutations or internal deletions, and alterations of downstream signaling components, as well as activation through autocrine feedback loops[18]. A key mechanism of post-translational control of RTK levels and signaling involves the regulation of their intracellular traffic. One aspect of RTK traffic that has received the most attention is their post-activation endocytic traffic into either lysosomal degradation or the alternative recycling pathway back to the plasma membrane, with the balance of these mechanisms a key determinant of the magnitude, duration, and type of cellular responses elicited by ligand-induced RTK activation[19]. Indeed, altered endocytic trafficking of RTKs, including the imbalance between recycling versus degradation, is now known to promote oncogenic signaling by RTKs[20,21].

To investigate the potential link of EHD1 to RTK-dependent tumorigenesis, we carried out studies using Ewing Sarcoma (EWS), the second most common malignant bone tumor in children and young adults[22], as a model. Despite advances in multimodality treatment strategies, the EWS prognosis remains poor, with cure rates below 25%, due to its aggressive and metastatic nature[23–25]. More than 85% of cases harbor reciprocal translocations that generate a currently undruggable fusion oncogene composed of portions of *EWS* and ETS transcription factor *FLI1*[24]. *EWS-FLI1* drives oncogenesis through altered transcriptional activity as well as other mechanisms that together promote a fully malignant phenotype[26,27].

Upregulation of signaling through multiple RTKs is implicated in EWS tumorigenesis, metastasis, and therapy resistance, with the role of insulin-like growth factor 1 receptor (IGF-1R) having received most attention[17]. IGF-1R was demonstrated to be required for EWS/FLI1-mediated transformation of EWS cells[28]. Furthermore, EWS/FLI1 and other EWS-associated fusion oncoproteins transcriptionally upregulate the IGF-1 expression[29]. EWS-FLI1 binding to IGF binding protein 3 (IGFBP-3) promoter was found to repress the expression of this key negative regulator of IGF-1R signaling, leading to constitutively active IGF-1R signaling in EWS cells[30]. IGF-1R and components of the IGF-1 receptor signaling pathway have also been associated with the development, progression, and metastasis of breast, non-small cell lung, and other solid cancers[31–33]. Many preclinical studies support the potential of IGF-1R targeting to limit tumorigenesis and metastasis[32,34,35]. In EWS in particular, IGF-1R inhibition has been explored[36–43] but the results of clinical trials with antibody- and tyrosine kinase inhibitor (TKI)-based IGF-1R targeting have been disappointing[44,45]. The inefficacy of IGF-1R targeting in the clinic likely reflects the lack of predictive markers of therapeutic response as well as our still incomplete understanding of the regulation of IGF-1R in tumors.

Given the important roles of IGF-1R and other RTKs in supporting the fusion oncoprotein-driven tumorigenesis and metastasis in EWS, we test our hypothesis that EHD1 overexpression enables high cell surface levels of RTKs as a novel pro-oncogenic mechanism using EWS as a model. Our results establish a critical positive role of EHD1 overexpression in EWS oncogenesis and demonstrate that EHD1-dependent endocytic recycling and pre-activation Golgi to the plasma membrane traffic of IGF-1R are essential for its oncogenic role.

## Results

**EHD1 is overexpressed in EWS patient tumors and correlates with shorter event-free and overall survival.** To assess if EHD1 is overexpressed in EWS patient tumors and if its overexpression bears any relationship with patient survival, we queried the publicly-available EWS patient tumor mRNA expression data using the R2 Genomics Analysis and Visualization Platform. Dichotomization of EHD1 mRNA expression levels into EHD1-High and EHD1-Low groups (mRNA expression cutoff: 439.8 TPM for event-free and 490.8 TPM for overall survival) followed by Kaplan–Meier survival analysis revealed that high EHD1 mRNA overexpression correlated with shorter event-free and overall survival in EWS patients (Fig. 1a, b). To assess if EHD1 expression is detectable in EWS patient tumors at the protein level, we carried out an immunohistochemistry (IHC) analysis of EHD1 expression in a tissue microarray of 324 EWS patient tumors. 88.6% of the 307 evaluable samples showed high EHD1 expression (IHC staining intensity of 2 or 3), while 7.49% showed low EHD1 expression (staining intensity of 1) with 3.91% deemed as negative (staining intensity of 0) (Fig. 1c, d). The level of EHD1 expression was significantly higher in metastases vs. the primary tumors (Fig. 1e). While limited survival data disallowed survival analyses, the IHC data further supported the idea that high EHD1 expression is a feature of a majority of EWS patient tumors. Overall, these analyses supported a potential pro-oncogenic role of EHD1 in EWS.

**EHD1 is required for the maintenance of in vitro pro-tumorigenic and pro-metastatic oncogenic traits of EWS cell lines.** To identify EWS cell models suitable for delineating the role of EHD1 in tumor biology, we first queried the Cancer Cell Line Encyclopedia (CCLE)/DepMap mRNA expression database. Most of the 19 included EWS cell lines showed moderate EHD1

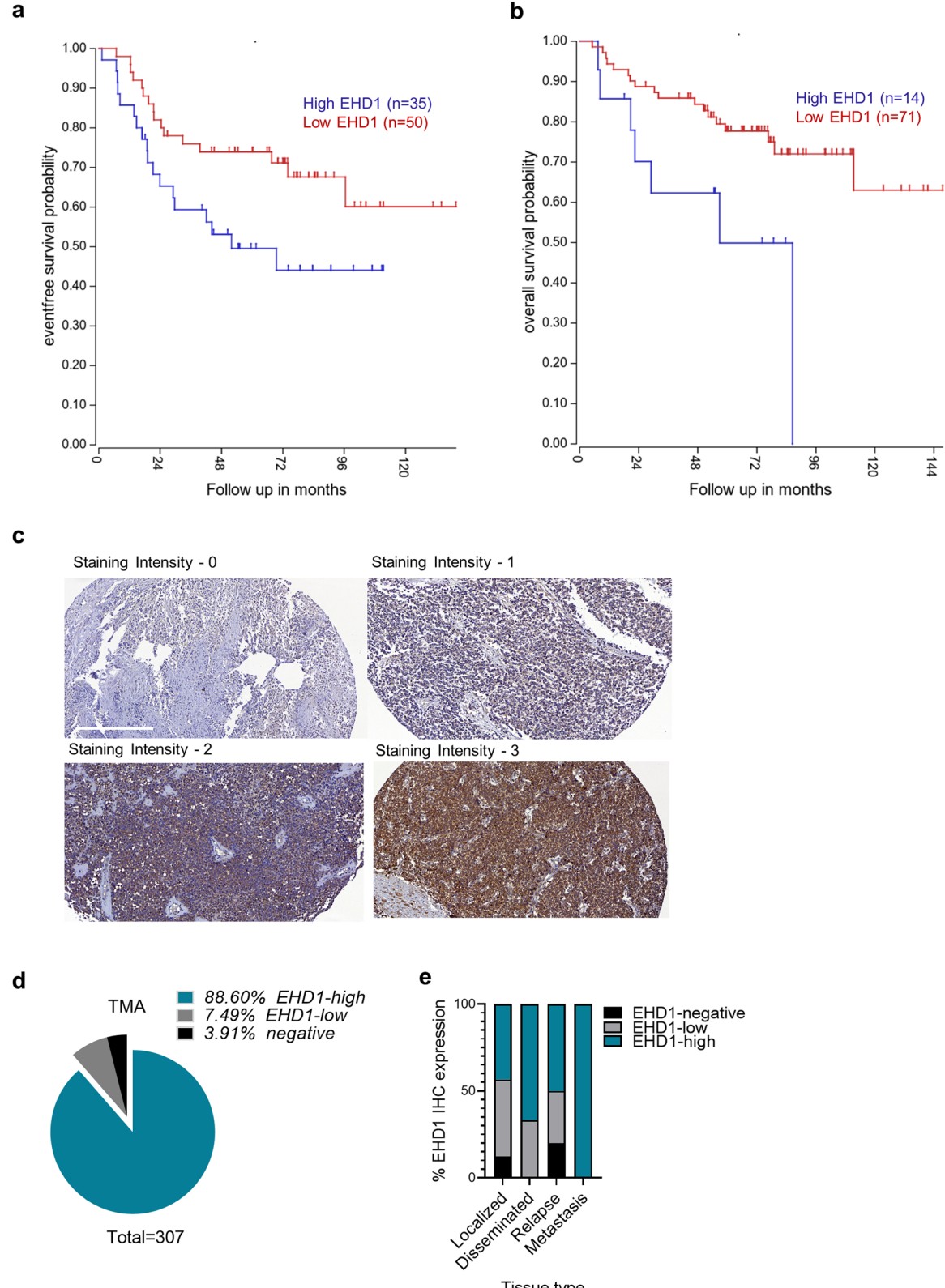

mRNA levels relative to levels across the total cell line panel (Supplementary Table 1). Analysis of a subset of EWS cell lines representing the three EWS-FLI1 fusion oncogene types (TC71 and A673—Type I EWS-FLI1 fusion; MHH-ES1, SK-ES-1—Type II EWS-FLI1 fusion; and A4573—Type III fusion) by immuno-blotting revealed a good correlation between mRNA and protein levels, with consistently lower EHD1 protein levels in SK-ES-1 compared to the other 4 EWS cell lines, which showed robust EHD1 expression (A4573 is absent in the CCLE data) (Supple-mentary Fig. S1). While EHD2 was undetectable, all cell lines showed EHD3 expression with variable levels of EHD4. Based on these results, we used lentiviral constructs to engineer TC71, A673, and SK-ES-1 cell lines stably expressing 3 distinct dox-ycycline (Dox)-inducible EHD1-specific shRNAs (shEHD1) or a

**Fig. 1 EHD1 is overexpressed in Ewing Sarcoma patient tumors and its overexpression is associated with shorter survival. a, b** Kaplan–Meier survival analysis of 85 EWS patients based on publicly-available EHD1 mRNA expression using the R2 Genomics Analysis and Visualization Platform. EHD1-high (blue); EHD1-low (red). Event-free survival analysis (**a**; $p = 0.038$) used a dichotomization cut-off of 439.8 (right panel), with $N = 35$ for EHD-high and $N = 50$ for EHD1-low group. Overall survival analysis (**b**; $p = 0.014$) used a dichotomization cut-off of 490.8 (right panel) with $N = 14$ for EHD1-high and $N = 71$ for EHD1-low groups. The dichotomization cutoffs represent the program-selected defaults based on statistical significance. **c, d** EHD1 overexpression in EWS patient primary tumor tissue microarrays examined by immunohistochemistry (IHC). **c** Representative examples of various intensities (on a scale of 0 to 3) of anti-EHD1 antibody staining; details in Methods. Scale bar, 300 μm. **d** Relative distribution of EHD1-high (staining intensity of 2 or 3), EHD1-low (staining intensity of 1), or EHD1-negative samples ($N = 307$). **e** Significantly higher expression of EHD1 in metastatic lesions as compared to localized disease, $\chi^2 = 22.389$; $p = 0.001$, Spearman's correlation coefficient $= 0.211$; $p < 0.001$.

non-targeting control shRNA (shNTC). The shEHD1 #2 and #3 lines with robust EHD1 knockdown (KD) specifically upon Dox treatment (Fig. 2a) were selected for further analyses.

First, we examined the impact of Dox-induced EHD1-KD on the various in vitro oncogenic traits. EHD1-KD markedly and significantly reduced the magnitude of cell proliferation, measured using the Cell-Titer Glo assay in TC71, A673, and SK-ES-1 cell lines (Fig. 2c, Supplementary Fig. S2a). Furthermore, EHD1-KD in A673 and TC71 cell lines induced a significant reduction in anchorage-independent growth on soft agar and tumorsphere forming ability (Fig. 2e, f, Supplementary Fig. S2b, c). EHD1-KD also induced a drastic reduction of transwell cell migration and invasiveness (migration through Matrigel) (Fig. 2g, h, Supplementary Fig. S2e–h). Treatment with the cell-proliferation inhibitor mitomycin-C excluded the role of reduced cell proliferation as a major contributor to reduction in cell migration and invasion; the modest reduction in proliferation upon mitomycin-C treatment at 24 h could not account for the nearly 85% reduction in migration and invasion ability (Supplementary Fig. S2d).

To further establish the pro-oncogenic role of EHD1 and its specificity, we generated CRISPR-Cas9 EHD1 knockout (KO) derivatives of TC71 and A673 cell lines and then used a lentiviral construct to stably express mouse Ehd1 (mEHD1) in the EHD1-KO cell lines to assess the rescue of any functional deficits (Fig. 2b, Supplementary Fig. S3). Indeed, EHD1-KO induced a pronounced decrease in cell proliferation and migratory abilities of both EWS cell lines, and this deficit was rescued by mEHD1 (Fig. 2d, i, Supplementary Fig. S2i); consistent with higher levels of the introduced mEHD1, the rescued cell lines displayed increased proliferation and migration relative to their wildtype parental lines. Further illustrating the pro-oncogenic role of EHD1 overexpression, introduction of mouse Ehd1 into EHD1-low SK-ES-1 cell line led to a marked and significant increase in cell proliferation, migration, and invasion compared to their parental cells (Fig. 2j–m, Supplementary Fig. S2j). RNA-seq analysis showed a marked reduction in cell cycle regulatory gene expression in Dox-treated shEHD1 vs. shNTC TC71 cell lines among the significantly downregulated pathways (Supplementary Fig. S4a–f) and qPCR analysis validated the downregulation of CDK4, CDK6, E2F1, E2F2, and PCNA mRNA levels (Supplementary Fig. S4d). Collectively, our KD, KO, rescue, and overexpression analyses strongly support a key positive role of EHD1 in promoting multiple pro-tumorigenic and pro-metastatic traits of EWS cells.

**EHD1 is required for in vivo EWS tumorigenesis.** To assess if the marked reduction in pro-oncogenic traits seen in vitro translates into impaired tumorigenesis in vivo, we implanted TC71 NTC, EHD1-KO, and mEHD1 rescue cell lines engineered with a lentiviral mCherry-enhanced luciferase reporter[46] in the tibias of Nude mice ($n = 8$ per group at the beginning) and monitored tumor growth by bioluminescence imaging. While the NTC tumors exhibited time-dependent growth (seen as an

increase in log10 photon flux), the EHD1-KO tumors failed to grow and, in fact, showed a reduction in photon flux; in contrast, implants of the mEHD1-rescued EHD1-KO cells exhibited rapid tumor growth with higher photon flux, and mice in this group reached the euthanasia endpoints a week earlier (Fig. 3a, b, Supplementary Fig. S5a, b). IVIS imaging of lungs resected at necropsy revealed detectable metastatic seeding in 3 of 8 mice implanted with NTC cells but in none of the mice implanted with EHD1-KO cells. In contrast, 7/8 mice implanted with mEHD1-rescued cells showed metastases (Fig. 3c, d). Notably, 1/8 NTC and 2/8 rescued cell line-implanted mice exhibited liver metastases. Morphometric analysis of tibial bone by micro-CT scanning showed reduced bone volume, trabecular number, thickness, and separation in mice implanted with NTC or mEHD1-rescued TC71 cells, indicative of increased tumor-induced bone degradation, with a significant amelioration of these defects in tibias of mice implanted with EHD1-KO cells (Fig. 3e, f).

To assess the impact of inducible EHD1 KD on pre-formed tumors, we implanted Nude mice with shNTC or shEHD1 (#3) TC71 cell lines carrying the TdTomato-luciferase reporter, and monitored the tumor growth by IVIS imaging, as above. Groups of tumor-implanted mice ($n = 7$/group for NTC and 6/group for shEHD1) were either followed as such or switched to Dox-containing water beginning at Day 10. Comparable time-dependent growth of shNTC TC71 implants without or with Dox treatment excluded any impact of Dox itself; in contrast, the growth of shEHD1 TC71 tumors was markedly reduced by Dox treatment compared to untreated mice ($p < 0.0001$; Supplementary Fig. S6a, b). Western blotting of resected tumor lysates confirmed the Dox-induced EHD1 KD in the shEHD1 group, and IHC staining with anti-human CD99 confirmed the tumor mass (Supplementary Fig. S6c, d). Tumors of Dox-treated shEHD1-implanted mice showed fewer proliferating tumor cells (Ki-67 staining) and an increase in apoptotic cells (cleaved-caspase3) (Supplementary Fig. S6e). Collectively, these results unequivocally demonstrate a requirement of EHD1 for EWS tumorigenesis and metastasis.

**Identification of IGF-1R as an EHD1 target in EWS.** Given our previous identification of EHD1 as a regulator of Golgi to plasma membrane traffic and subsequent signaling of EGFR and CSF1R[4,5], we hypothesized that regulation of RTKs may underlie the requirement of EHD1 in EWS oncogenesis. We, therefore, probed a phospho-RTK profiling array incorporating 49 of the 58 human RTKs with lysates of untreated (control) vs. Dox-treated (KD) shEHD1 #3 TC71 or A673 cell lines. The levels of phospho-IGF-1R were specifically reduced upon Dox treatment in both cell lines, while changes in other phospho-RTKs were not seen in either (Fig. 4a). Consistent with our findings with EGFR and CSF1R[5], analysis of TC71 cell lines harboring two distinct shRNAs (#2 or #3) demonstrated a reduction in total IGF-1R levels upon EHD1-KD (Fig. 4b). These results were further validated using control vs. CRISPR-KO TC71 and A673 cell lines;

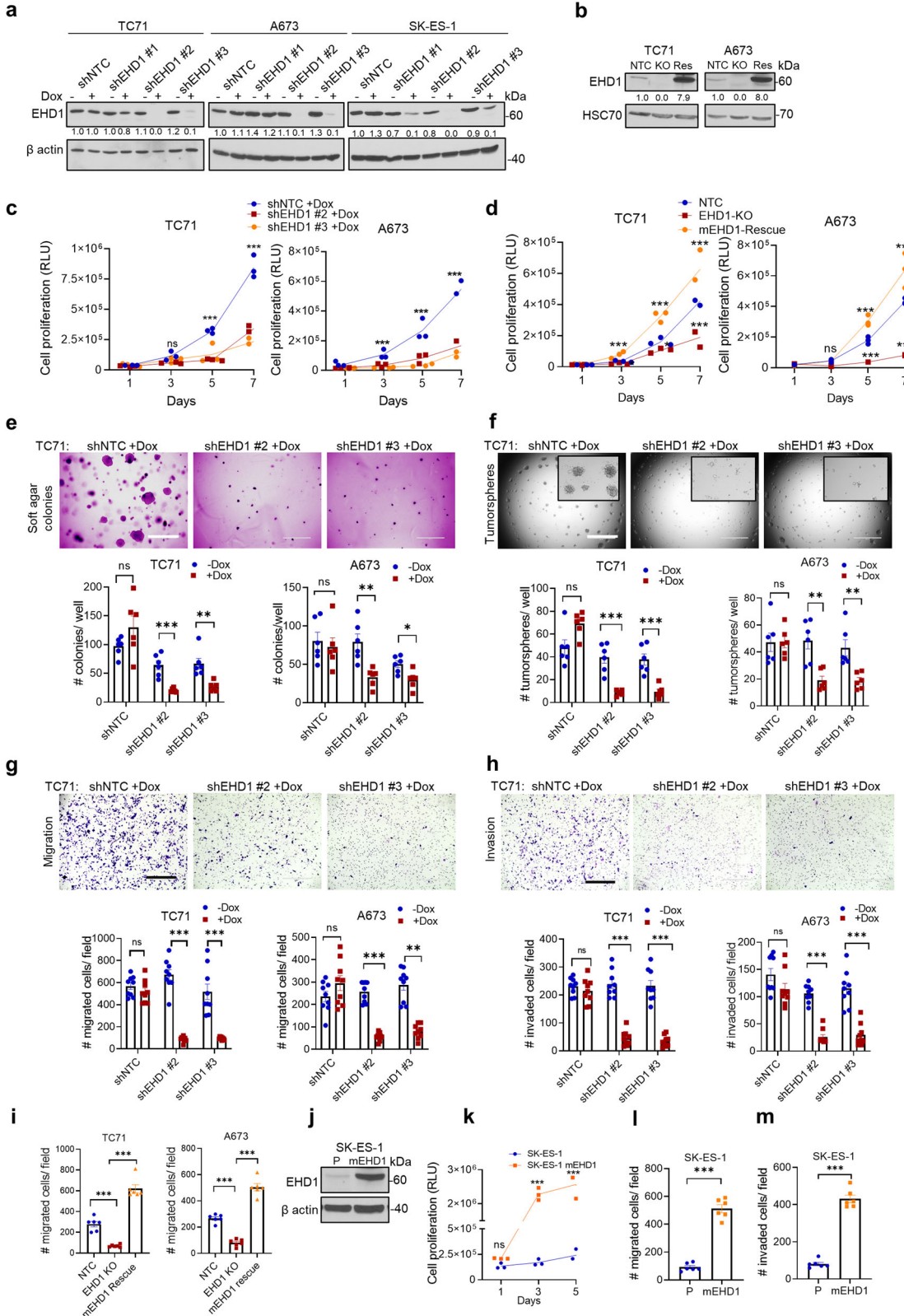

notably, mEHD1-rescued KO cell lines exhibited higher total IGF-1R levels than the non-targeted controls, correlating with higher mEHD1 levels compared to that of endogenous EHD1 in control cells (Fig. 4c). qPCR analyses demonstrated comparable IGF-1R mRNA levels between the NTC and EHD1-KO cell lines, excluding the EHD1 regulation of IGF-1R levels at the mRNA level (Supplementary Fig. S7a).

Notably, the predominant IGF-1R species in EHD1-KO TC71 and A673 cell lines engineered with mEHD1 re-expression and in EHD1-low SK-ES-1 cell lines engineered with mEHD1 over-expression was distinctly slower-migrating than that in parental wildtype cells (Fig. 4c). A survey of the literature[47,48] suggested the likelihood that this mobility shift may reflect differential N-linked glycosylation of IGF-1R. To test this possibility, we

**Fig. 2 EHD1 is required to sustain the in vitro oncogenic traits of Ewing Sarcoma cell lines. a** Western blot analysis of Doxycycline (Dox)-inducible knockdown of EHD1 in the indicated EWS cell lines. Cells stably expressing the non-targeting control shRNA (shNTC) or EHD1-specific shRNAs (shEHD1 #1, 2, or 3) were grown for 72 h without (−) or with (+) 0.5 μg/ml Dox before lysis and immunoblotting. β-actin served as a loading control. **b** Western blot analysis of CRISPR-Cas9 based EHD1-KO in EWS cell lines and their derivatives with mouse EHD1 (mEHD1) expression. The indicated cell lines engineered with non-targeting control (NTC) or EHD1-targeted Cas9-sgRNA (KO) two-in-one constructs or the KO lines with mEHD1 rescue (Res) were analyzed for EHD1 expression. HSC70 served as a loading control. **c** Impaired cell proliferation upon EHD1 knockdown. The indicated shNTC and shEHD1 TC71 or A673 cell lines pre-treated with Dox for 48 h were plated in 96-well plates and cell proliferation assessed at the indicated time points using the Cell-Titer-Glo assay. Y-axis, Relative Luminescence Units (RLU) as a measure of increase in the number of viable cells. Data points represent mean +/− SEM of three experiments, each with six replicates. **d** Impaired cell proliferation upon EHD1 knockout (KO) and rescue of proliferation defect by mEHD1. Cell proliferation was assessed as in (**c**). **e** Impaired soft agar colony formation upon EHD1 knockdown. The indicated shNTC and shEHD1 TC71 or A673 cell lines pre-treated with Dox for 48 h were plated in soft agar and the colony numbers quantified after 3 weeks of culture in the presence of Dox. Top, representative images of TC71 cells; scale bar, 1000 μm. Bottom, mean +/− SEM of two experiments each in triplicates. **f** Impaired tumorsphere formation upon EHD1 knockdown. Top, Representative images of TC71 cells; scale bar, 1000 μm. Bottom, Mean +/− SEM of two experiments each in triplicates. **g** Impaired trans-well cell migration upon EHD1 knockdown. Top, representative images of TC71 cells; scale bar, 400 μm. Bottom, quantification of the number of migrated cells per high-power field; mean +/− SEM of three experiments each in triplicates. **h** Impaired invasion through Matrigel-coated trans-wells upon EHD1 knockdown. Top, representative images of TC71 cells; scale bar, 400 μm. Bottom, quantification of the number of invaded cells per high-power field; Mean +/− SEM of three experiments each in triplicates. **i** Impaired transwell cell migration upon EHD1 knockout (KO) and rescue of migration defect by mEHD1. Analyses done as in (**g**). **j** Immunoblot analysis demonstrating mEHD1 overexpression relative to endogenous EHD1 in parental cells (P) in SK-ES-1 cells. **k** Increased cell proliferation of SK-ES-1 cells upon mouse EHD1 (mEHD1) overexpression by Cell-Titer-Glo assay. Data points represent mean +/− SEM of 3 experiments each with six replicates. **l, m** Transwell migration and invasion assays in SK-ES-1-mEHD1 cells as compared to control cells. Data points represent mean +/− SEM of two experiments each in triplicates; *p < 0.05, **p < 0.01, ***p < 0.001, ns = not significant.

treated the EWS cell line lysates with PNGase-F for enzymatic removal of N-linked oligosaccharides. In all mEHD1-overexpressing cell lines, the up-shifted IGF-1R band shifted down and co-migrated with the predominant band seen in parental wildtype cell lines (Fig. 4d, e, Supplementary Fig. S7b). Treatment with Lambda phosphatase did not affect the mobility of the up-shifted IGF-1R bands (Supplementary Fig. S7c, d), ruling out a role for altered phosphorylation as a cause of the IGF-1R mobility shift in EHD1-rescue/overexpression cell lines. Thus, EHD1 overexpression appears to promote more complete or additional N-linked glycosylation of IGF-1R.

The cell surface levels of RTKs determine their access to ligands and hence the downstream responses[20]. To further assess the impact of EHD1-KO on cell surface levels of IGF-1R, we carried out live-cell IGF-1R immunostaining followed by FACS analysis on control vs. EHD1-KO TC71 and A673 cell lines under three distinct conditions: 1. Cells cultured in regular medium with 10% FBS (steady state). 2. Cells in regular medium treated with IGF-1 (100 ng/ml) to promote ligand-induced internalization and degradation of IGF-1R. 3. Cells in regular medium treated with IGF-1 (100 ng/ml) for 16 h to promote the downregulation of cell surface IGF-1R followed by culture in low serum (0.5%) medium without added IGF-1 for 24 h to allow the newly-synthesized receptor to accumulate at the cell surface. The cell surface IGF-1R on control cells decreased upon IGF-1 treatment followed by an increase upon culture in low-serum/IGF-1-free medium, reflecting the transport of newly synthesized IGF-1R to the cell surface (Fig. 4f–h). The EHD1-KO cells, in contrast, exhibited lower cell surface levels under all conditions, and the extent of IGF-1-induced surface IGF-1R downregulation was smaller than in control cells (Fig. 4f–h). Concurrent immunoblotting confirmed the lower IGF-1R levels in KO cells under all conditions examined (Fig. 4i). Since FACS analyses above were done on trypsin-EDTA released cells, we carried out additional experiments to ensure that the target protein recognized by the anti-IGF-1R antibody used (which recognizes an IGF-1R epitope within aa 283–440 in the α-chain that includes the L1 domain) remained intact under these conditions. Immunoblotting with an IGF-1R α-chain-specific antibody revealed that the α-chain was indeed intact in trypsin-EDTA treated cells and the signals were comparable to those in directly lysed cells (Supplementary Fig. S7e). Immunofluorescence microscopy further confirmed

the lower cell surface IGF-1R levels in EHD1-KO compared to control TC71 or A673 cell lines (Supplementary Fig. S7f).

As a complementary approach to immunofluorescence-based approaches above to assess the role of EHD1 in cell surface IGF-1R expression, we carried out live cell surface biotin labeling using a non-cell-permeable cross-linker followed by IGF-1R immuno-precipitation and streptavidin blotting to selectively quantify the cell surface IGF-1R pool[49]. Indeed, we found reduced levels of biotinylated IGF-1R signals in EHD1-KO compared to their wildtype parental cell lines, while robust biotinylated IGF-1R signals were seen in mEHD1-rescued TC71 or A673 cell lines (Fig. 4j, upper panels). Anti-IGF-1R immunoblotting confirmed the expected IGF-1R immunoprecipitation pattern (Fig. 4j, lower panels). Collectively, our results strongly support the conclusion that EHD1 is a major determinant of the cell surface expression of IGF-1R in EWS cells. Notably, anti-IGF-1R IHC of the EWS patient TMAs (same as those used for EHD1 staining) showed that 60.35% of the 227 interpretable samples exhibited high (staining intensity of 2-3) IGF-1R staining (Fig. 4k–n), with a positive correlation (Spearman's Correlation Coefficient = 0.179) between EHD1 and IGF-1R staining (Fig. 4k–n, Supplementary Tables 2–7).

**EHD1 controls the cell surface levels of IGF-1R by regulating its intracellular traffic itinerary.** EHD1 is known to facilitate recycling of many non-RTK receptors following their endocytosis via the Rab11+ endocytic recycling compartment[50] but whether EHD1 regulates RTK recycling, a key mechanism to counteract the alternate lysosomal delivery and degradation after ligand-induced internalization[20], is unknown. Consistent with EHD1-dependent RTK recycling, we previously observed that EHD1 colocalizes with a constitutively active oncogenic mutant or wildtype EGFR in endocytic compartments[5]. Furthermore, ecto-pically overexpressed IGF-1R and EHD1 were shown to co-immunoprecipitate (co-IP), partially in an IGF-1 dependent manner, and to colocalize in intracellular vesicular compartments post-IGF-1 stimulation[51].

To test the role of EHD1 in regulating the itinerary of pre-existing cell surface IGF-1R, we first carried out co-IP analyses of endogenous IGF-1R and EHD1 in lysates of TC71 and A673 cells that were serum/IGF-1-deprived for 24 h and then left unstimu-lated or stimulated with IGF-1 (50 ng/ml) for 1 h. EHD1/IGF-1R

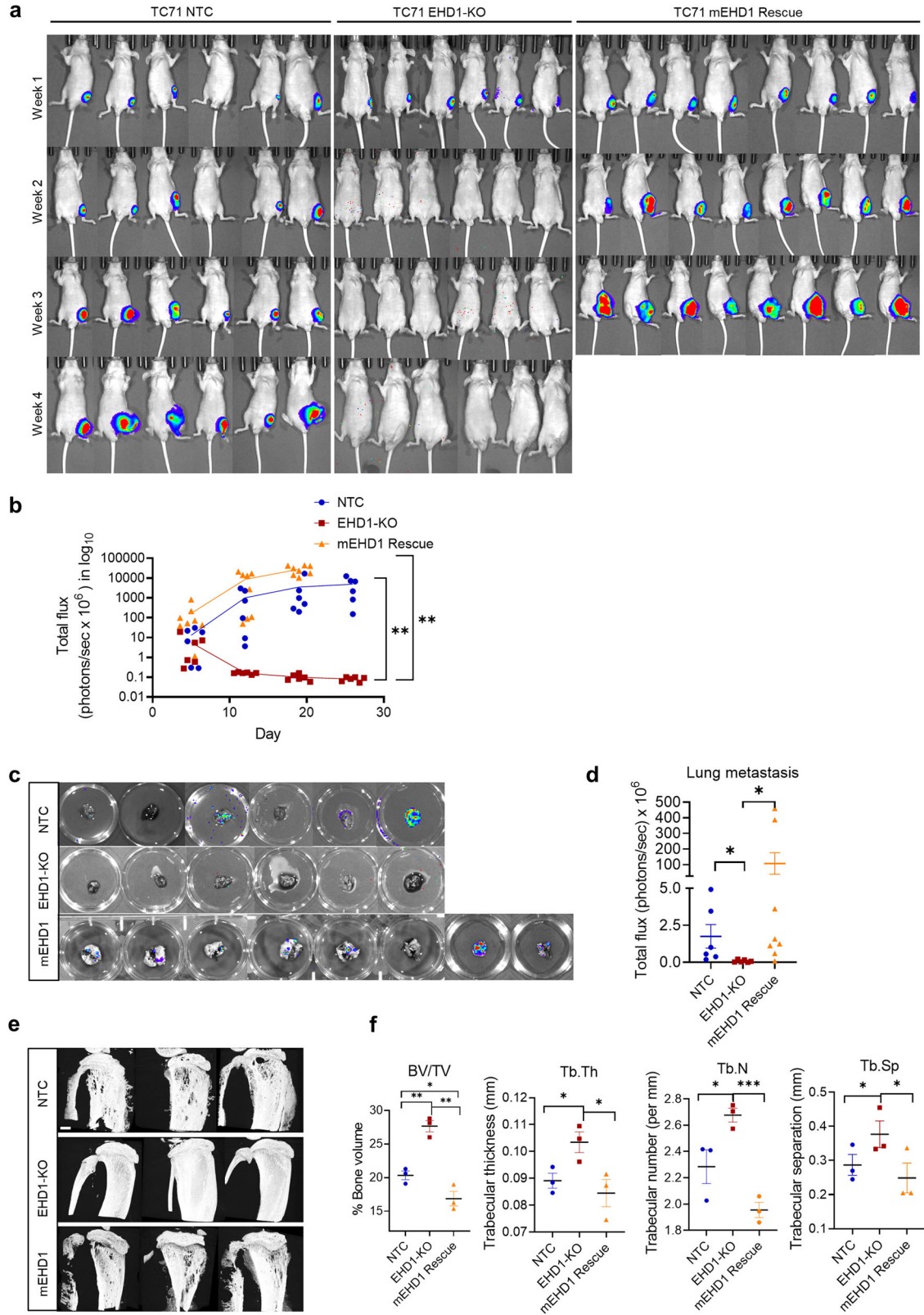

complexes were seen both under unstimulated and IGF-stimulated conditions (Fig. 5a). Confocal imaging demonstrated that most IGF-1R was localized at the cell surface after serum/IGF-1 starvation, with a small intracellular pool colocalizing with EHD1; upon IGF-1 stimulation, a significantly larger intracellular, presumably endosome-localized, pool of IGF-1R colocalized with EHD1 (Supplementary Fig. S8a, b). To assess if the intracellular

colocalization of EHD1-IGF-1R reflects a role of EHD1 in endocytic recycling of cell surface IGF-1R, serum/IGF-deprived (starved) control or EHD1-KO EWS cell lines were treated with cycloheximide (CHX) to inhibit further protein synthesis and pulsed with IGF-1 to promote IGF-1R endocytosis followed by chase in IGF-1-free medium for various times. Confocal imaging demonstrated that internalized IGF-1R became colocalized with

**Fig. 3 Loss of EHD1 expression markedly impairs the growth and metastasis to lungs of bone-implanted EWS cells.** $2 \times 10^5$ TC71 cells edited with non-targeting (NTC) or EHD1-targeted sgRNA (EHD1-KO), or the EHD1-KO cells rescued with mEHD1, all carrying a mCherry-luciferase reporter were injected in tibias of 6-week-old nude mice (8/group) and primary tumor growth was monitored by bioluminescence imaging at the indicated time points in mice with detectable bioluminescent signals at the outset (6 for NTC and EHD1-KO groups; 8 for Rescue group). **a** Images of individual mice with superimposed luminescence signals over time. **b** Plots of log total flux values over time. Differences between groups were analyzed using two-way ANOVA; $**p < 0.01$. **c, d** Bioluminescence signals of lungs harvested at necropsy are shown as individual images (**c**) and as quantified total flux (**d**). **e** Micro-CT scanned images of tibias isolated from mice in the indicated groups. 3 mice per group were scanned. Scale bar, 300 μm. **f** Quantification of percent bone volume (BV/TV), trabecular thickness (Tb.Th), trabecular number (Tb.N), and trabecular separation (Tb.Sp) of scanned images from (**e**), by CTAn software. Data represent Mean $+/-$ SEM ($*p < 0.05$, $**p < 0.01$, $***p < 0.001$).

the endocytic recycling compartment marker RAB11 in control cells (0 min chase) but subsequently (30- and 60-min chase) reappeared at the cell surface with a decrease in the RAB11-colocalizing intracellular signal, indicating efficient recycling; in contrast, EHD1-KO cells showed continued IGF-1R/RAB11 colocalization during chase concurrent with reduced cell surface levels (Fig. 5b, Supplementary Fig S9a). Co-staining with a plasma-membrane marker, wheat-germ agglutinin (WGA), to specifically mark the cell surface pool of IGF-1R confirmed the reduced cell surface IGF-1R levels in EHD1-KO TC71 and A673 cells at 30- and 60-min chase times following IGF-1 induced receptor internalization as compared to control cells (Supplementary Figs. S10–11). These results support the role of EHD1-dependent endocytic recycling as one mechanism by which it sustains the cell surface levels of IGF-1R.

To assess if EHD1 also functions as a positive regulator of the Golgi to cell surface transport of newly synthesized IGF-1R, as we reported with CSF1R and EGFR[4,5], we first treated TC71 or A673 cell lines with IGF-1 to maximally deplete the cell surface and total IGF-1R (due to ligand-induced degradation). We then switched the cells to serum/IGF-1-deprivation medium and used confocal imaging to assess the appearance of newly synthesized IGF-1R in the Golgi compartment (co-staining with the Golgi marker GM130) and at the cell surface, with quantification of the latter. At time zero (after switching to serum/IGF-1-deprivation medium), both control and EHD1-KO cells exhibited weak overall and cell surface IGF-1R signals; the cell surface IGF-1R staining progressively increased in control cells with a minor intracellular pool colocalizing with GM130 (Fig. 5c, Supplementary Fig. S9b). In contrast, only a minor increase in the cell surface pool of IGF-1R was observed over time in EHD1-KO cells; on the other hand, the KO cells exhibited strong intracellular IGF-1R persistently localizing in the GM130+ Golgi compartment (Fig. 5c, Supplementary Fig. S9b).

The marked decrease in the cell surface and total IGF-1R levels in EHD1-depleted cells, without any change in IGF-1R mRNA levels, suggested that IGF-1R is targeted for degradation. Based on our findings with CSF1R in bone marrow-derived macrophages[4], we assessed if this reflected the mistargeting of IGF-1R to the lysosomes upon EHD1 depletion. Treatment of steady-state cultures of Control and EHD1-KO EWS cell lines with Bafilomycin-A1 (Baf-A1), a lysosomal proton pump blocker, led to a dramatic recovery of the total IGF-1R levels in EHD1-KO cells, nearly approaching the levels in untreated or Baf-A1-treated control EWS cells; Baf-A1 treatment had an insignificant effect on IGF-1R levels in control cells (Fig. 6a, b). Consistent with the WB findings, confocal imaging revealed that while the pool of IGF-1R localized to LAMP1+ lysosomes in control cells was relatively unchanged upon Baf-A1 treatment, a marked and significant increase in this pool was evident in Baf-A1-treated vs. untreated EHD1-KO EWS cells (Fig. 6c–f). Collectively, these results suggest that EHD1 is required for efficient transport of IGF-1R from the Golgi and the endosomal recycling compartment to the plasma membrane and that loss of EHD1 results in mistargeting of the cell surface-destined IGF-1R to the lysosome for degradation.

**IGF-1R signaling is required for EHD1 to promote the oncogenic behavior of EWS cells.** Since optimal cell surface expression is essential for ligand-induced activation of RTKs[18], and IGF-1R activation is critical for it to promote oncogenesis and metastasis[52], we postulated that the positive role of EHD1 to promote the oncogenic behavior of EWS cells reflects the enhancement of IGF-1R signaling. Indeed, while control TC71 or A673 cells exhibited robust and relatively sustained IGF-1-induced phosphorylation of IGF-1R itself and of nodal readouts of its downstream signaling through AKT and MAPK signaling pathways (phospho-AKT-Ser473 and phospho-ERK1/2-Thr202/Tyr204), these responses were drastically and significantly impaired in EHD1-KO EWS cells (Fig. 7a, b). Notably, the mEHD1 expressing TC71 and A673 EHD1-KO cell lines showed a rescue of the IGF-1-induced phosphorylation of IGF-1R, AKT, and ERK compared to that in EHD1-KO cell lines (Fig. 7c, d). Moreover, treatment with IGF-1R inhibitor Linsitinib exhibited similar IGF-1-induced signaling defects as in EHD1-KO TC71 and A673 cell lines (Fig. 7e, f). Gene-set enrichment analysis (GSEA) of the RNA-seq data of control vs. Dox-induced shEHD1 EWS cell lines showed a significant enrichment for genes involved in PI3K-AKT-mTOR signaling, further supporting the premise that EHD1 regulates IGF-1R signaling to promote oncogenesis (Fig. 7g). Indeed, flow cytometric analysis of annexin-V/propidium iodide (PI) co-stained cells revealed a significantly higher proportion of apoptotic cells in EHD1-KO EWS cell lines; IGF-1R inhibitor Linsitinib significantly increased the proportion of early and late apoptotic cells in control EWS cells and more so in EHD1-KO TC71 and A673 cells (Fig. 7h, i). The additional Linsitinib-induced inhibition of the IGF-1-induced oncogenic traits in EHD1-KO cell lines is consistent with residual low levels of IGF-1R in these cells. Moreover, IGF-1-dependent cell proliferation and migration were drastically and significantly reduced in EHD1-KO TC71 and A673 cell lines compared to their controls (Fig. 7j–l). Furthermore, Linsitinib significantly reduced the IGF-1-induced proliferation and migration of control EWS cell lines, and the combination of EHD1-KO and Linsitinib produced an even greater reduction in these responses (Fig. 7j–l).

To directly assess the requirement of IGF-1R for EHD1-dependent elevation of the oncogenic behavior of EWS cells, we first targeted the IGF-1R using multiple IGF-1R loss of function approaches in the mEHD1-overexpressing SK-ES-1 cell line, which exhibits a specific EHD1 overexpression-dependent enhancement of oncogenic traits. First, we assessed the impact of negating IGF-1R signaling, either by treatment with the IGF-1R inhibitor Linsitinib or by transient siRNA-mediated knockdown of IGF-1R on the oncogenic phenotypes of these cells. Treatment with Linsitinib eliminated the elevated induction of phosphorylation on IGF-1R, AKT, and ERK1/2 seen in mEHD1-overexpressing vs. parental SK-ES-1 cells (Fig. 8a). We confirmed that IGF-1R but not the control siRNA transfection produced an effective and specific KD of IGF-1R in SK-ES-1-mEHD1 cells (Fig. 8b). Importantly, treatment with Linsitinib as well as IGF-1R siRNA transfection

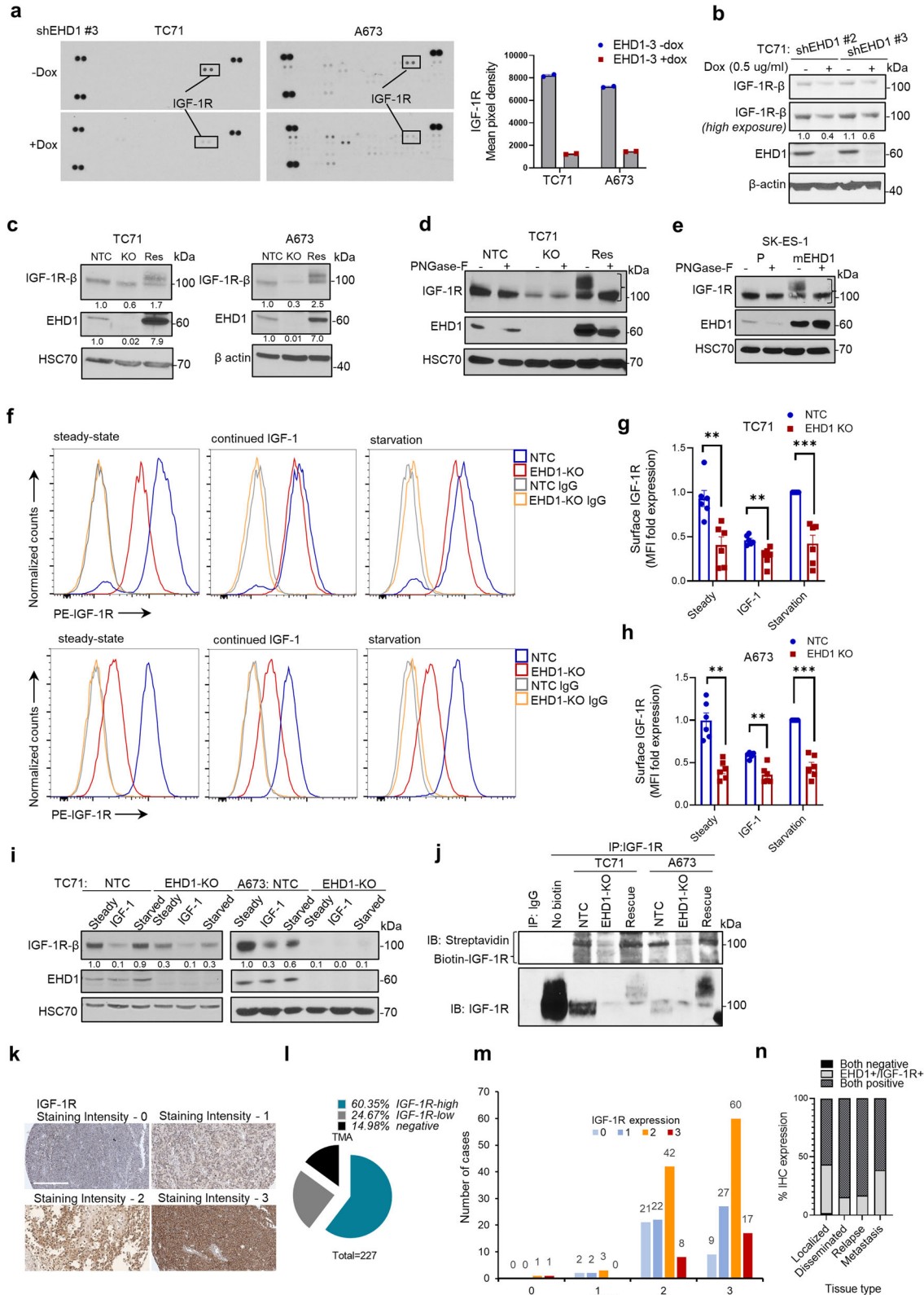

significantly and comparably reduced the elevated levels of cell proliferation in SK-ES-1-mEHD1 cells while control siRNA transfection had no impact (Fig. 8c). Further, while mEHD1-overexpressing cells exhibited a significantly lower level of apoptosis, IGF-1R siRNA or its inhibition with Linsitinib elevated the proportion of apoptotic cells above that seen in parental cells while control siRNA had no effect (Fig. 8d, Supplementary

Fig. S12a). Analysis of the IGF-1-induced cell migration and invasion demonstrated that both the siRNA KD of IGF-1R and pharmacological inhibition with Linsitinib reversed the EHD1-overexpression-dependent elevation of these oncogenic traits in SK-ES-1 cells (Fig. 8e, f, Supplementary Fig. S12b, c). As an independent inhibitory approach, we assessed the impact of treatment with an IGF-1R inhibitory monoclonal antibody

**Fig. 4 Identification of insulin-like growth factor-1 receptor (IGF-1R) as a regulatory target of EHD1 in EWS. a** Phospho-RTK antibody array analysis. Membranes arrayed with antibodies against phosphorylated versions of 49 human RTKs (each in duplicate) were probed with lysates of TC71-shEHD1 or A673-shEHD1 treated with or without Dox. Left: Images of membranes with IGF-1R spots indicated. Right: Densitometric quantification of IGF-1R signals. **b** Western blot showing reduced total IGF-1R protein levels in TC71 cells upon Dox-induced EHD1 knockdown. Cell lysates were probed with an anti-IGF-1Rβ antibody with β-actin as a loading control. **c** Reduction in total IGF-1R levels upon EHD1-KO and rescue by mEHD1 expression in EHD1-KO EWS cells. Lysates of the indicated cell lines probed with anti-IGF-1Rβ antibody; HSC70 or β-actin served as loading controls. **d, e** Slower migration of IGF-1R band in immunoblots of exogenous mEHD1 rescue/overexpressing EWS cell lines TC71 (**d**) and SK-ES-1 (**e**) is due to increased N-linked glycosylation. Indicated cell lines were left untreated or treated with PNGase-F (10,000 U/ml for 30 min) followed by anti-IGF-1R immunoblotting. HSC70 served as loading controls. **f–h** EHD1-KO leads to reduced cell surface expression of IGF-1R on EWS cell lines. Control and EHD1-KO TC71 (top panel) and A673 (bottom panel) cells were grown in regular medium (steady-state), stimulated with IGF-1 (100 ng/ml) for 16 h prior to analysis to promote the IGF-1R degradation (continued IGF-1), or cells pre-treated with IGF-1 were switched to low serum-containing and IGF-1-free medium (starvation) to promote the cell surface accumulation of newly-synthesized IGF-1R. Live cells were stained with anti-IGF-1R or IgG control antibody and analyzed by FACS. **f** Representative histograms. **g, h** Quantification of surface IGF-1R exprssion. Data represents the fold ratio of median fluorescence intensity (MFI) relative to NTC cells under starvation condition (assigned a normalized value of 1). Mean $+/-$ SEM of six independent experiments. ($*p < 0.05$, $**p < 0.01$, $***p < 0.001$, ns = not significant). **i** Representative immunoblotting (with densitometric quantification) for total IGF-1R expression in samples analyzed under (**f**). **j** Reduced cell surface IGF-1R levels in EHD1-KO TC71 and A673 cell lines and rescue by mEHD1. 500 µg aliquots of lysate protein of live-cell surface biotinylated cell lines were subjected to IGF-1R immunoprecipitation followed by blotting with Streptavidin (top; for biotin-labeled IGF-1R) and IGF-1R (bottom; for total IGF-1R). **k–n** Positive correlation of EHD1 and IGF-1R expression in EWS patient tumors. Anti-IGF-1R IHC staining was carried out on TMAs from the same patient cohort as that analyzed for EHD1 expression (in Fig. 1). **k** The representative examples of the IGF-1R staining intensity of 0–3, Scale bar, 300 µm. **l** shows the relative distribution of high (staining intensity of 2–3; 60.35%), low (staining intensity of 1; 24.67%) or negative (staining intensity of 0; 14.9%) IGF-1R staining among 227 evaluable patients. **m** The correlation between EHD1 and IGF-1R staining intensities. Y-axis, number of cases displaying IGF-1R staining intensities of 0,1, 2, or 3. X-axis, EHD1 staining intensities, 0–3. Spearman's Correlation Coefficient = 0.179, $p = 0.009$. **n** Expression of EHD1 and IGF-1R in localized disease, disseminated, relapse, and metastatic lesions.

1H7[53,54]. FACS-based titration indicated that 4 mg/ml 1H7 was saturating for SK-ES-1-mEHD1 cells (Supplementary Fig. S12d, e). Inclusion of 1H7 in the assay demonstrated a marked inhibition of the elevated IGF-1-induced cell proliferation, migration, and invasion phenotypes of SK-ES-1-mEHD1 cells vs. their parental EHD1-low cells while the control IgG had no significant effect (Fig. 8g–i, Supplementary Fig. S12f, g). Collectively, these loss-of-function analyses demonstrated that upregulation of IGF-1R levels and signaling are necessary for the oncogenic phenotypes imparted on EWS cells by EHD1 overexpression. Finally, in a gain-of-function genetic approach, we assessed if exogenous overexpression of IGF-1R could restore the defective oncogenic phenotypes of EHD1-KO EWS cell lines. We stably expressed a GFP-tagged IGF-1R in EHD1-KO TC71 or A673 cell lines and demonstrated robust overexpression of the GFP-IGF-1R by anti-IGF-1R immunoblotting (recognizable due to its larger molecular size) (Supplementary Fig. S12h, i). GFP-IGF-1R overexpressing cells, but not the vector-transfected cells, exhibited a restoration of the IGF-1 induced phosphorylation of AKT and ERK1/2, with robust induction of phosphorylation of GFP-IGF-1R itself (Fig. 8j). Importantly, GFP-IGF-1R overexpressing EHD1-KO TC71 and A673 cell lines exhibited robust IGF-1 induced cell proliferation, migration, and invasion comparable to their non-KO parental cell lines (Fig. 8k–p). Thus, multiple loss of function approaches combined with a gain of function genetic approach provide compelling evidence that upregulation of IGF-1R signaling constitutes a major pathway by which EHD1 overexpression unleashes EWS oncogenesis. Of note, cell lines with mEHD1 overexpression (Figs. 7c, d and 8a) as well as EHD1-KO cell lines with GFP-IGF-1R overexpression (Fig. 8j) exhibited higher basal pERK1/2, which was reversible upon longer (48 h) starvation in SK-ES-1 system (Fig. 8a) (but could not be tested in TC71 and A673 models due to lower survival after longer starvation) as well as by Linsitinib (Fig. 8a). These results further support a key role of IGF-1R-dependent signaling in EHD1-dependent oncogenic phenotype.

## Discussion

Besides driver oncogenes, tumor cells turn on multiple adaptive pathways for successful primary tumor growth and metastasis. Delineating these oncogenesis-enabling pathways is likely to identify novel biomarkers of malignant behavior and therapeutic responses of tumors and, in some cases, offer opportunities for therapeutic targeting. Here, using Ewing Sarcoma (EWS) as a tumor model, we demonstrate that the intracellular vesicular traffic regulatory protein EHD1 promotes tumorigenesis and metastasis by serving as a required element of IGF-1R traffic to enable IGF-1R-mediated oncogenic programs. While EWS is a relatively uncommon malignancy, it is the second most common bone and soft tissue tumor of children and young adults[22]. Importantly, the novel mechanistic insights we uncover using EWS models are likely to be broadly relevant to malignancies where RTKs serve as drivers or enablers of oncogenesis and EHD1 protein is overexpressed.

In a large EWS tumor panel, we found moderate to high EHD1 overexpression in nearly 90% of patients, with significantly higher levels in metastatic tumors (Fig. 1d, e). Query of publicly available data revealed the high EHD1 mRNA expression to be associated with shorter patient survival (Fig. 1a, b). Thus, clinical data support a positive role of EHD1 protein in EWS tumorigenesis. These findings are consistent with reports of EHD1 over-expression in other cancers, in many cases associated with shorter patient survival or resistance to therapy[8,11,16].

Our comprehensive genetic analyses of EWS cell models definitively demonstrate that EHD1 propels tumorigenic and metastatic behavior in EWS. Use of Doxycycline-inducible shRNA knockdown in cell line models demonstrated a strong dependence of cell proliferation, tumorsphere growth, cell migration, and invasion on EHD1 (Fig. 2c–h), with a stronger impact in cells lines with higher EHD1 expression (A673 and TC71) and a more modest impact in cells (SK-ES-1) with lower EHD1 levels (Supplementary Fig. S2a, g, h). Reciprocally, ectopic mouse Ehd1 overexpression in the latter cells markedly enhanced their pro-tumorigenic and pro-metastatic traits (Fig. 2j–m). EHD1-KO in A673 and TC71 cell models confirmed the requirement of EHD1 for the in vitro pro-tumorigenic and pro-metastatic behavior of EWS cells, and re-expression of mEHD1 restored the EHD1-KO defects (Fig. 2i). Use of Dox-inducible KD or EHD1-KO EWS cell models in a bone implant model in nude mice demonstrated a key role of EHD1 in EWS tumorigenesis and metastasis in vivo, and the defective tumorigenic ability of EHD1-KO cells was

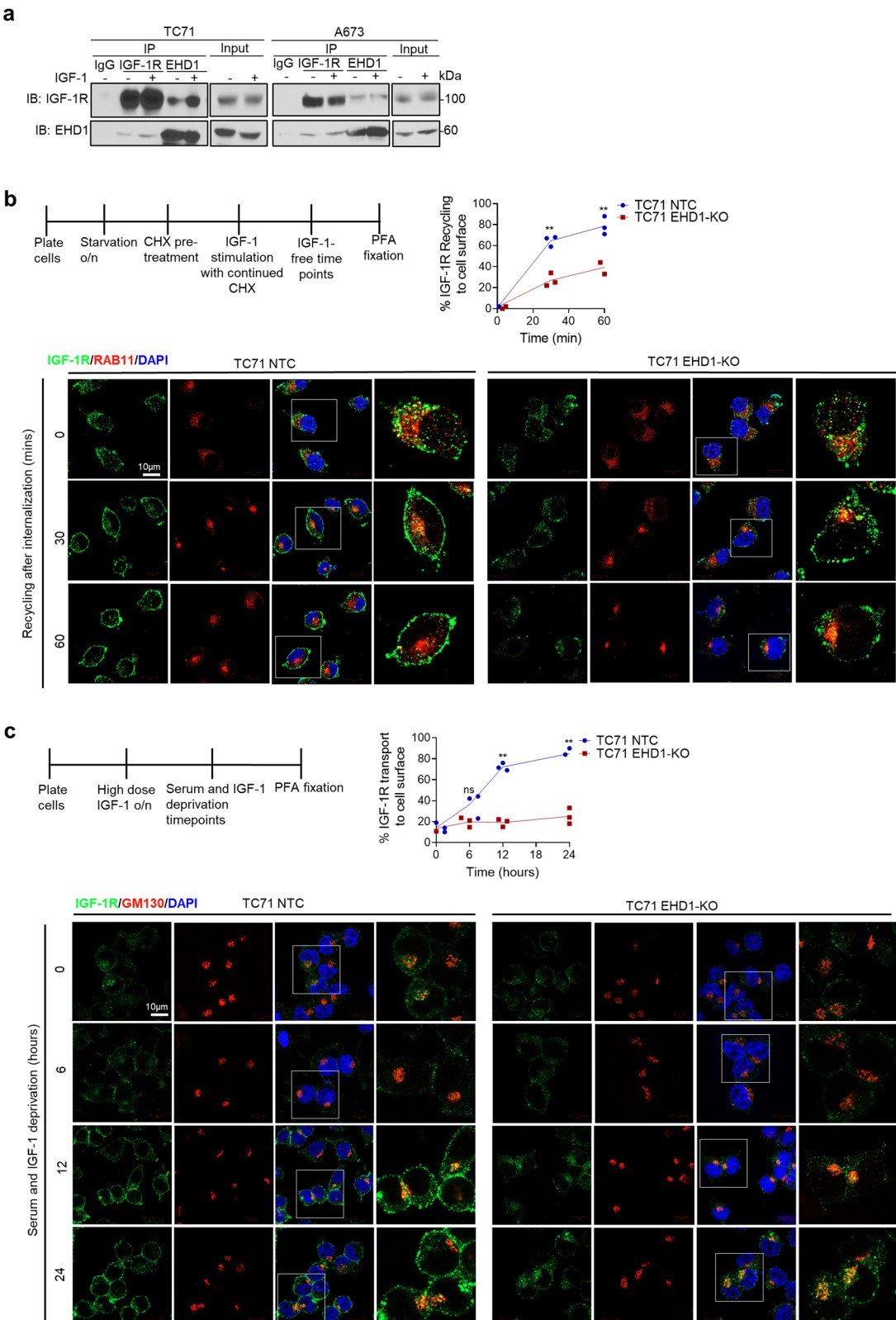

completely restored by mEHD1 rescue (Fig. 3a, b). Furthermore, the modest metastases forming ability of parental EWS cells was completely abolished by EHD1-KO; notably, the mEHD1-rescued EHD1-KO cells, which express higher EHD1 levels than the parental cells, showed significantly more metastatic growths (Fig. 3c, d). A hallmark of bone-associated tumors is the destruction of the surrounding bone[55]. Indeed, compared to significant bone destruction by parental EWS cell implants, EHD1-KO cells failed to do so, and the process was accentuated in mEHD1-rescued KO cells (Fig. 3e, f). Collectively, our clinicopathological studies combined with our in vitro and in vivo genetic perturbation studies provide compelling evidence for a key role of EHD1 overexpression in sustaining EWS tumorigenesis and metastasis.

**Fig. 5 EHD1 controls cell surface IGF-1R levels by regulating its endocytic recycling and Golgi to plasma membrane traffic. a** EHD1-IGF-1R association in EWS cells. Anti-IGF-1Rβ or anti-EHD1 antibody immunoprecipitates (IP) from 1 mg lysate protein aliquots of the indicated cell lines were subjected to Western blotting for IGF-1Rβ or EHD1; co-IP is observed in both directions. **b** EHD1-KO impairs IGF-1R endocytic recycling. TC71 NTC or EHD1-KO cells pretreated with cycloheximide (50 μg/ml) for 2 h to prevent new protein synthesis were treated with IGF-1 to promote the ligand-induced IGF-1R internalization (time 0), followed by incubation in IGF-1-free medium (30 and 60 min). Fixed and permeabilized cells were co-stained for IGF-1Rβ (green), RAB11 (recycling endosome marker; red) and nuclei (DAPI, blue), and analyzed using confocal imaging to assess the delivery of IGF-1R into recycling endosomes and its subsequent recycling to the cell surface. Top left, a schematic of the treatments. Bottom, Co-staining for IGF-1R and RAB11. The zoomed in panels (4th columns for each cell line) show high co-localization of IGF-1R and Rab11+ in TC71-NTC cells at time 0 (after IGF-1-induced internalization) with reduction over time, concurrent with increased plasma membrane IGF-1R signals. In EHD1-KO cells, a more persistent co-localization is seen over time with lesser increase in plasma membrane signals over time. Top right, the data is expressed as a % of fluorescence intensity of plasma membrane IGF-1R using ImageJ; details in Methods. **c** EHD1-KO impairs the Golgi to plasma membrane traffic of IGF-1R. The TC71-NTC and EHD1-KO cells pre-treated with IGF-1 (100 ng/ml) for 16 h to deplete the cell surface IGF-1R (time 0) were subjected to serum/IGF-1 deprivation for 6, 12, or 24 h. Fixed and permeabilized cells were co-stained for IGF-1Rβ (green), GM130 (Golgi marker; red) and nuclei (DAPI, blue), and analyzed using confocal imaging to assess the delivery of newly-synthesized IGF-1R at the Golgi followed by its delivery to the plasma membrane. Top left, a schematic of the treatments. Bottom, Co-staining for IGF-1R and GM130. The zoomed in panels (4th columns for each cell line) show a small GM130-colocalizing pool of IGF-1R in TC71-NTC cells with time-dependent increase in its cell surface pool. EHD1-KO cells show an increase in the GM130-colocalizing pool of IGF-1R over time with essentially no increase in the cell surface IGF-1R. Top right, quantification of the percentage of IGF-1R fluorescence signals at the plasma membrane using ImageJ; details in Methods. 80 cells were analyzed from three independent experiments. **b**, **c** Scale bar, 10 μm. Mean +/− SEM (*$p < 0.05$, **$p < 0.01$, ns = not significant).

To confirm that the impact of loss of EHD1 on pro-metastatic traits such as migration and invasion and consequently on the in vivo metastatic abilities of EWS cell lines was not simply due to marked reduction in cell proliferation, we conducted the migration and invasion assays in the presence of a mitotic inhibitor, mitomycin C. These analyses demonstrated that the modest reduction in cell proliferation observed at short time points used for these assays cannot account for the strong impairment of the KO EWS cell line migration and invasion (Supplementary Fig. S2d). The strong impact of EHD1-KO or re-expression/overexpression on metastasis as well as bone destruction is consistent with EHD1 overexpression promoting multiple pro-oncogenic behaviors in EWS.

Our studies provide novel insights into how EHD1 serves in a pro-tumorigenic and pro-metastatic role. Our mechanistic studies were focused on two key considerations, one the established role of EHD1 in regulating intracellular traffic of multiple cell surface receptors[1,56,57], and our previous studies that have established a key role of EHD1 to ensure high cell surface expression of RTKs by regulating critical aspects of their traffic[4,5]. Our unbiased query of human receptor tyrosine kinome identified IGF-1R as a specific target (Fig. 4a). Our comprehensive cell biological analyses demonstrate that EHD1 is required for Golgi to plasma membrane traffic of newly-synthesized IGF-1R to ensure high pre-activation levels of total and cell surface IGF-1R (Fig. 5c), the latter a requirement for subsequent ligand-induced activation of signaling and cellular responses[32]. In addition, EHD1 plays a positive role in post-activation recycling of IGF-1R to help return it to the cell surface (Fig. 5b), uncovering a second trafficking mechanism known to help sustain cell surface RTK levels by countering their lysosomal targeting[20]. Consistent with the key roles of EHD1 in regulating IGF-1R traffic to sustain its cell surface expression while negating its lysosomal degradation, our biochemical and subcellular localization analyses establish that lack of EHD1 leads to marked mistargeting of IGF-1R to lysosomes where it is degraded (Fig. 6). Previous analyses have shown that EHD1 can interact with IGF-1R[51], which we find is also the case in EWS cell models (Fig. 5a), but a role for EHD1 in regulating IGF-1R traffic had not been shown previously. Thus, our studies establish a novel role for EHD1 in regulating IGF-1R traffic from intracellular vesicular compartments (Golgi and recycling endosomes) to the cell surface.

In the context of the EHD1 promotion of IGF-1R traffic from the Golgi to the cell surface, it is notable that EHD1-KO TC71 and A673 cell lines re-expressing mEHD1 as well as mEHD1

overexpressing SK-ES-1 (EHD1-low) cell line exhibited a predominant slower-migrating IGF-1R band in western blots compared to that seen in their parental cells. Elimination of this mobility difference by PNGase-F treatment established that elevation of EHD1 levels led to higher N-linked glycosylation of IGF-1R (Fig. 4d, e, Supplementary Fig. S7b). This is notable since lack of N-linked glycosylation of IGF-1R was previously found to impair the membrane localization of IGF-1R and led to anti-IGF-1R antibody (figitumumab) insensitivity in gastric and hepatocellular cancer cell lines[47]. Studies in Ewing Sarcoma cell lines have also shown that inhibition of N-linked glycosylation of IGF-1R downregulated the plasma membrane bound IGF-1R and consequently decreased the IGF-1R signaling and EWS cell survival[48]. These studies have reported a positive role of N-linked glycosylation of IGF-1R in its oncogenicity[47,58]. Further studies are therefore warranted to assess whether the promotion of N-liked glycosylation of IGF-1R and potentially other RTKs[4,5] is mechanistically linked to the EHD1-dependent enhancement of their Golgi to cell surface traffic.

Notably, ligand-induced internalization, lysosomal degradation, and recycling of IGF-1R are well-established aspects of its traffic and signaling[59–62]. Post-endocytic recycling of IGF-1R has been shown to be positively regulated by myoferlin[63], RAB11-FIP3[64], and GIGYF1[65]. Thus, our studies identify EHD1 as a new regulator of IGF-1R endocytic recycling. RAB11-FIP3 is a component of endocytic recycling, in which EHD1 plays a key role[1], and a family member RAB11-FIP2 interacts with EHD proteins[50], suggesting the possibility that EHD1 may function together with RAB11-FIP proteins to regulate the recycling of IGF-1R and potentially other RTKs. Interestingly, ligand-dependent IGF-1R localization to Golgi has been associated with the migratory behavior of tumor cells, suggesting signaling capabilities of the Golgi-localized receptor[66]. In previous studies, we found EHD1 to play a role in retrograde traffic of the cell surface EGFR to Golgi[5], suggesting the possibility that EHD1 could play a similar role in IGF-1R traffic.

In contrast to its post-activation traffic, mechanisms that regulate the availability of IGF-1R at the cell surface prior to ligand binding have been less explored. Interestingly, Smoothened protein was found to positively regulate IGF-1R levels in lymphoma and breast cancer cell lines by stabilizing it in plasma membrane lipid rafts and preventing its lysosomal targeting[67]. Whether Smoothened regulates endocytic recycling or Golgi to cell surface IGF-1R traffic was not explored. Notably, we have shown that EHD1 regulates the traffic of Smoothened in primary cilia[68],

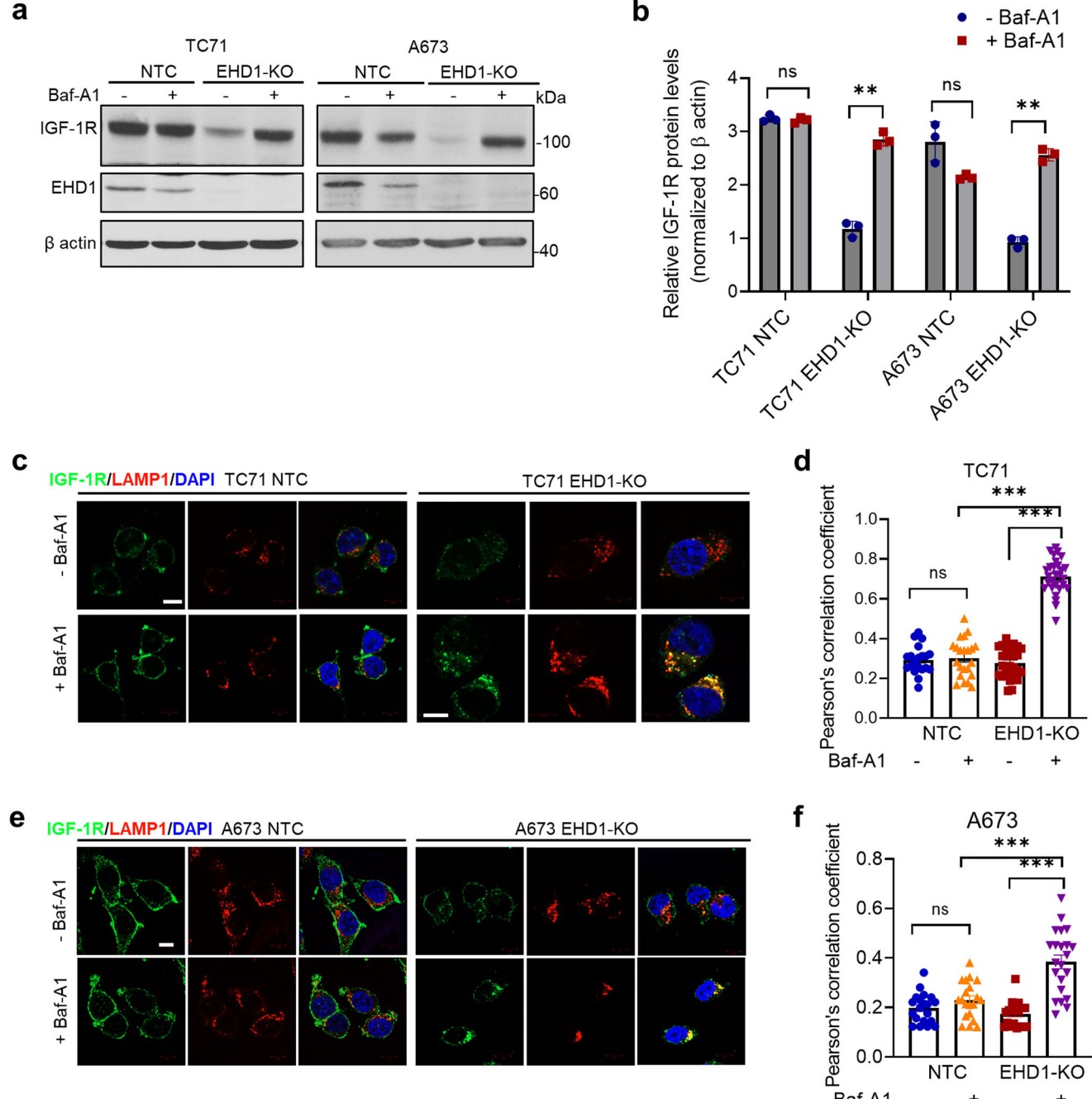

**Fig. 6 Loss of EHD1 expression leads to lysosomal degradation of IGF-1R. a, b** Recovery of IGF-1R protein levels upon inhibition of lysosomal protein degradation with Bafilomycin-A1. NTC or EHD1-KO TC71 and A673 cell lines were switched to low serum/IGF-1 free medium for 6 h in the absence or presence of Bafilomycin-A1 (200 nM) and total IGF-1R levels in cell lysates were analyzed by western blotting (**a**). The quantified IGF-1R signals normalized to β-actin loading control are shown in (**b**). Data represent mean +/− SEM of 3 experiments. Note the significant increase (**p < 0.01) in IGF-1R levels in EHD1-KO cells with no significant change in NTC cells (ns, not significant). **c–f** Lysosomal mistargeting of IGF-1R in EHD1-KO EWS cells. NTC or EHD1-KO TC71 and A673 cells were left untreated or treated with bafilomycin-A1 as in (**a, b**) and co-stained for IGF-1R (green), LAMP1 (lysosome marker, red) and nuclei (DAPI, blue). IGF-1R localization to lysosomes (yellow) is visualized in merged images (third columns) in the representative images shown in (**c**) and (**e**). Scale bar, 10 μm. Pearson's correlation coefficients (**d, f**) of the co-localized IGF-1R and LAMP1 fluorescence signals were determined from analyses of n > 20 cells per group from three independent experiments (**p < 0.01, ***p < 0.001, ns = not significant).

raising the possibility that Smoothened and EHD1 may co-regulate IGF-1R traffic.

Our findings linking EHD1 overexpression to regulation of an RTK that is well-established to control multiple aspects of oncogenesis provided a plausible basis for EHD1's pro-oncogenic role we uncovered. We provide multiple lines of evidence that this indeed is the case. Reduced cell surface IGF-1R expression upon

EHD1-KO directly translated into reduced activation of downstream signaling (Fig. 7a–f), and transcriptomic analyses support this conclusion (Fig. 7g). Thus, our analyses clearly establish that EHD1 overexpression, by sustaining elevated levels of total and cell surface IGF-1R, promotes multiple aspects of oncogenesis in EWS. While signaling through IGF-1R is well-established to promote oncogenesis in EWS[35], we directly establish that

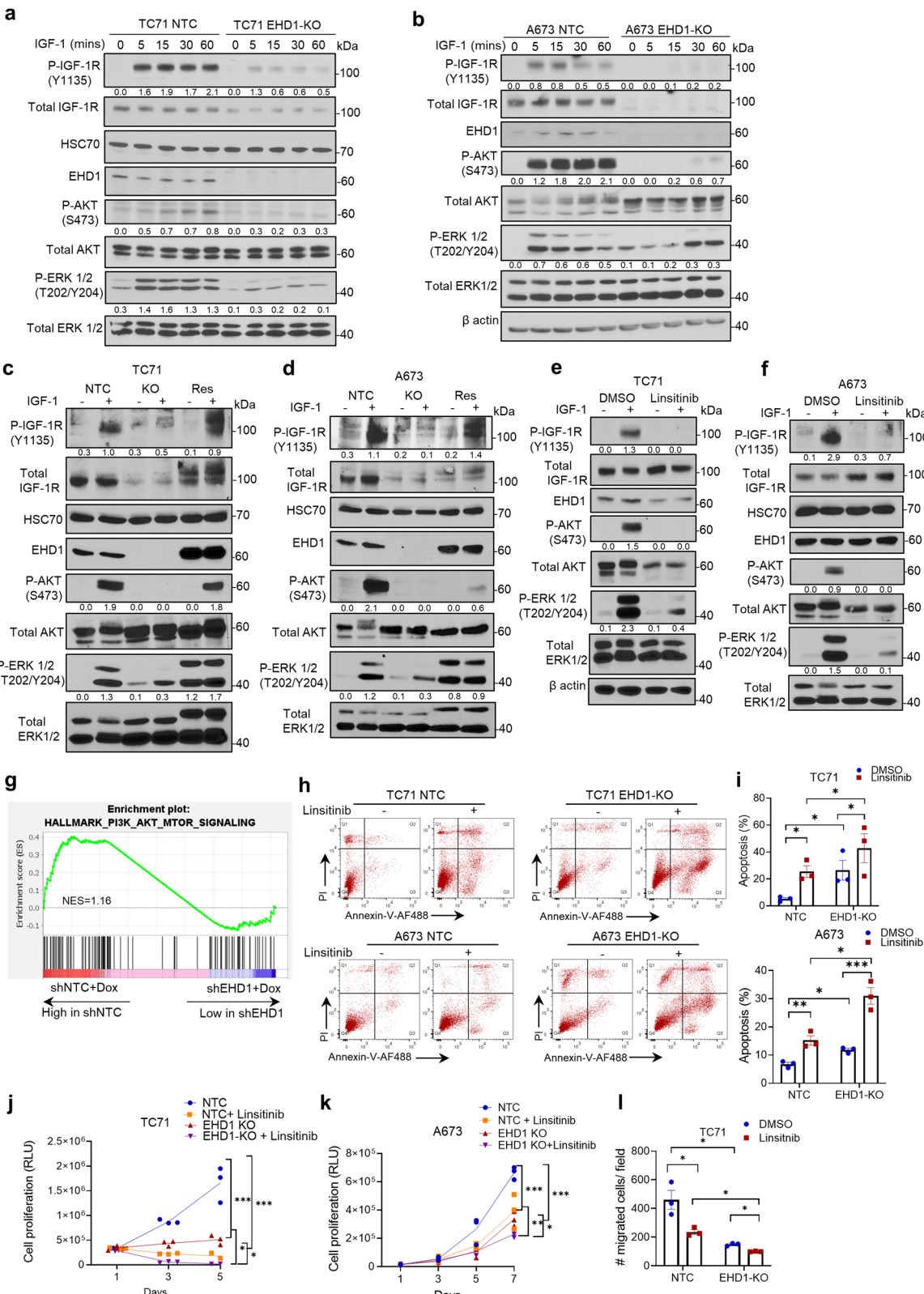

elevation of IGF-1R levels and subsequent IGF-1R-mediated signaling underlies the ability of EHD1 to promote the oncogenic behavior of EWS cells. Using the mEHD1-overexpressing SK-ES-1 cell model of EHD1-driven elevation of oncogenic behavior, our multi-pronged studies using siRNA KD, kinase inhibition, and an inhibitory antibody approach demonstrates a requirement of IGF-1R for EHD1 overexpression-driven oncogenic traits (Fig. 8a–i, Supplementary Fig. S12a–g). Conversely, overexpression of GFP-IGF-1R in EHD1-KO EWS cell lines rescued their oncogenic attributes (Fig. 8j–p). Thus, our studies clearly establish the upregulation of IGF-1R levels and signaling by overexpressed EHD1 as a key oncogenic adaptation in EWS. Consistent with this conclusion, analysis of a large cohort of EWS patient samples showed a significant positive correlation between EHD1 and IGF-

**Fig. 7 Loss of EHD1 expression in EWS cells impairs the IGF-1-dependent signaling downstream of IGF-1R. a, b** Western blot analysis of phosphorylation of IGF-1R and key signaling pathway reporters (AKT and ERK1/2). NTC or EHD1-KO TC71 and A673 cell lines were pre-starved for 24 h in low serum/IGF-free medium and left unstimulated (0) or were stimulated with IGF-1 (50 ng/ml) for the indicated time points (minutes). Cell lysates were analyzed by Western blotting with the indicated antibodies, with β-actin or HSC70 as loading control. Densitometric quantification of the phosphorylation signals of IGF-1R, AKT, and ERK normalized to the values of corresponding total protein signals are indicated below each panel. **c, d** Western blot analysis of phosphorylation of IGF-1R and key signaling pathway reporters (AKT and ERK1/2) in mEHD1 Rescued TC71 and A673 cell lines as compared to NTC and EHD1-KO cell lines. Cells were treated as in "**a, b**" above and either left unstimulated or stimulated with IGF-1 (50 ng/ml) for 30 min. **e, f** TC71 and A673 cell lines were pre-starved for 24 h in low serum/IGF-free medium combined with treatment with either DMSO or 1 μM Linsitinib and either left unstimulated or stimulated with IGF-1 (50 ng/ml) for 30 min. Cell lysates were analyzed by Western blotting with the indicated antibodies, with β-actin or HSC70 as loading control. Densitometric quantification of the phosphorylation signals of IGF-1R, AKT and ERK normalized to the values of the corresponding total protein signals are indicated below each panel. **g** Gene-set enrichment analysis (GSEA) from RNA-sequencing of two groups of TC71 cell lines—TC71 shEHD1+Dox vs. shNTC+Dox, shows enrichment of PI3K-AKT-mTOR signaling genes in shNTC+Dox cells and significant downregulation of the same in the shEHD1+Dox group. **h, i** EHD1-KO impairs the IGF-1-dependent pro-survival effects in EWS cells. Flow cytometric analysis of apoptosis in the indicated cells treated with or without 1 μM linsitinib for 24 h as assessed by Annexin-V and PI staining. Representative flow panel with the indicated treatments (**h**). **j, k** Impaired IGF-1-induced proliferation in EHD1-KO EWS cell lines. NTC or EHD1-KO TC71 and A673 cells were cultured in regular medium for 24 h, switched to medium with 1% FBS and 100 ng/ml IGF-1 in the absence or presence of 1 μM IGF-1R inhibitor linsitinib and cell proliferation measured at the indicated time points by Cell-Titer Glo assay. Data represent mean +/− SEM of three experiments, each in six replicates. **l** Impaired IGF-1-induced cell migration in TC71 cell line. NTC or EHD1-KO TC71 cells plated in top chambers of trans-wells in the absence or presence of 1 μM IGF-1R inhibitor linsitinib, with migration towards the medium with 1% FBS and 100 ng/ml IGF-1 in lower chambers. Data represents mean +/− SEM of three experiments, *$p < 0.05$, **$p < 0.01$, ***$p < 0.001$.

1R protein levels (Fig. 4k–n). Notably, while ~90% of EWS tumors exhibited moderate/high EHD1 expression (Fig. 1d), only ~60% of them exhibited high IGF-1R levels (Fig. 4l). One plausible explanation for this discrepancy is that EHD1 may upregulate other RTKs in cases where IGF-1R levels are not elevated, consistent with EHD1 regulation of other RTKs[4,5,11,16]. Notably, while IGF-1R signaling is altered in most cases of EWS[69], aberrations of other RTKs are also found[17]. As another plausible reason for the discrepancy, EWS-FLI-dependent aberrations in IGF-1R are multifactorial, including the upregulation of ligands or downregulation of inhibitory components[27,30], and such factors may predominate in patients where IGF-1R levels themselves are not upregulated. Broader RTK analyses combined with EHD1 expression studies as well as concurrent analyses of multiple components of IGF-1R signaling network should help address these possibilities.

Our studies demonstrating a requirement of EHD1 for IGF-1R cell surface traffic and signaling as a key component for its requirement for tumorigenic and metastatic behaviors of EWS cell models raise the question of whether EHD1 might be a suitable target in EWS since the impact of IGF-1R targeting has been low to modest[31,34,37,44]. Given the EHD1 regulation of IGF-1R and other RTKs[4,5] as well as its targeting of additional pathways shown in other cancers[9,11,12,14–16,70], it appears likely that EHD1 targeting may still be viable and may potentially be combined with IGF-1R targeting. Additionally, the cell surface levels of RTKs not only dictate the levels of ligand-induced and kinase-dependent signaling as documented for IGF-1R in this study, but also dictate their kinase-independent signaling as demonstrated for several RTKs including EGFR and IGF-1R[71–76]. Such kinase-independent signaling has been linked to kinase inhibitor resistance[77]. Also, the expression of other RTKs provides a prevalent mechanism of resistance to RTK-targeted therapies[78,79], and the ability of EHD1 to target these could be an advantage. While detailed studies are needed to test this speculative model, the additive effects of linsitinib treatment and EHD1-KO in EWS cell lines on apoptosis or cell migration readouts (Fig. 7h–l) are consistent with such an idea. Notably, in non-small cell lung cancer cell models, elevated levels of EHD1 correlated with insensitivity to EGFR inhibition and such insensitivity was overcome by genetic depletion of EHD1[16].

Our findings using an EWS model have potential implications for the pro-oncogenic role of EHD1 and RTK-dependent sustenance of tumorigenesis and metastasis in other cancers. EHD1 overexpression is linked to shorter survival and chemotherapy/EGFR-TKI resistance in NSCLC[16], apparently through PI3K-AKT pathway activation as a result of EHD1 interaction with the microtubule protein TUBB3 and stabilization of microtubules[16] and through promotion of aerobic glycolysis via a 14-3-3z-dependent b-catenin-c-Myc activation pathway[14]. While these mechanisms may operate independently of RTK signaling, the key roles of the wildtype or mutant EGFR as well as IGF-1R and other RTKs in NSCLC pathogenesis and therapeutic resistance[80] raise the possibility that EHD1 overexpression may activate these pathways by sustaining RTKs, as we show in the EWS models. Association of EHD1 overexpression with EGFR-TKI resistance in NSCLC[11,16] and with higher expression of EGFR, phospho-EGFR, and RAB11-FIP3[12] support this idea.

While our studies focus on the linkage of EHD1 with an RTK, EHD1 overexpression may also regulate other oncogenesis-related cell surface receptors, given its broader roles. Indeed, EHD1 overexpression was shown to promote cancer stem cell-like traits in glioblastoma and lung cancer by promoting CD44 recycling while suppressing its degradation[15,70], promote cisplatin resistance in NSCLC by regulating cisplatin accumulation in cells presumably by regulating transporter levels[9], and potentiate angiogenesis by promoting b2 adrenergic receptor recycling[13]. Cell biological studies have also shown a positive role of EHD1 in β1 integrin recycling[56]. Future studies of the kind described here in the context of an RTK, IGF-1R, should help uncover the individual or combined roles of the various EHD1-regulated cell surface receptors in promoting tumorigenesis and metastasis.

In conclusion, our analyses in an EWS tumor model show that EHD1 overexpression promotes oncogenesis by post-translationally upregulating the trafficking itinerary of an RTK, IGF-1R (Fig. 9).

## Methods

**Ewing sarcoma patient tissue microarrays and immunohistochemical analysis.** A total of 324 paraffin-embedded samples from ESFT (Ewing Sarcoma Family of Tumors) patients from the period between April 1971 and May 2007 treated at Istituto Ortopedico Rizzoli (IOR), Bologna, Italy, and at the Department of Pathology of the University of Valencia Estudi General (UVEG), Spain were analyzed within the context of two European Translational Research projects [PROTHETS (http://www.prothets.org) and EuroBo-Net (http://www.

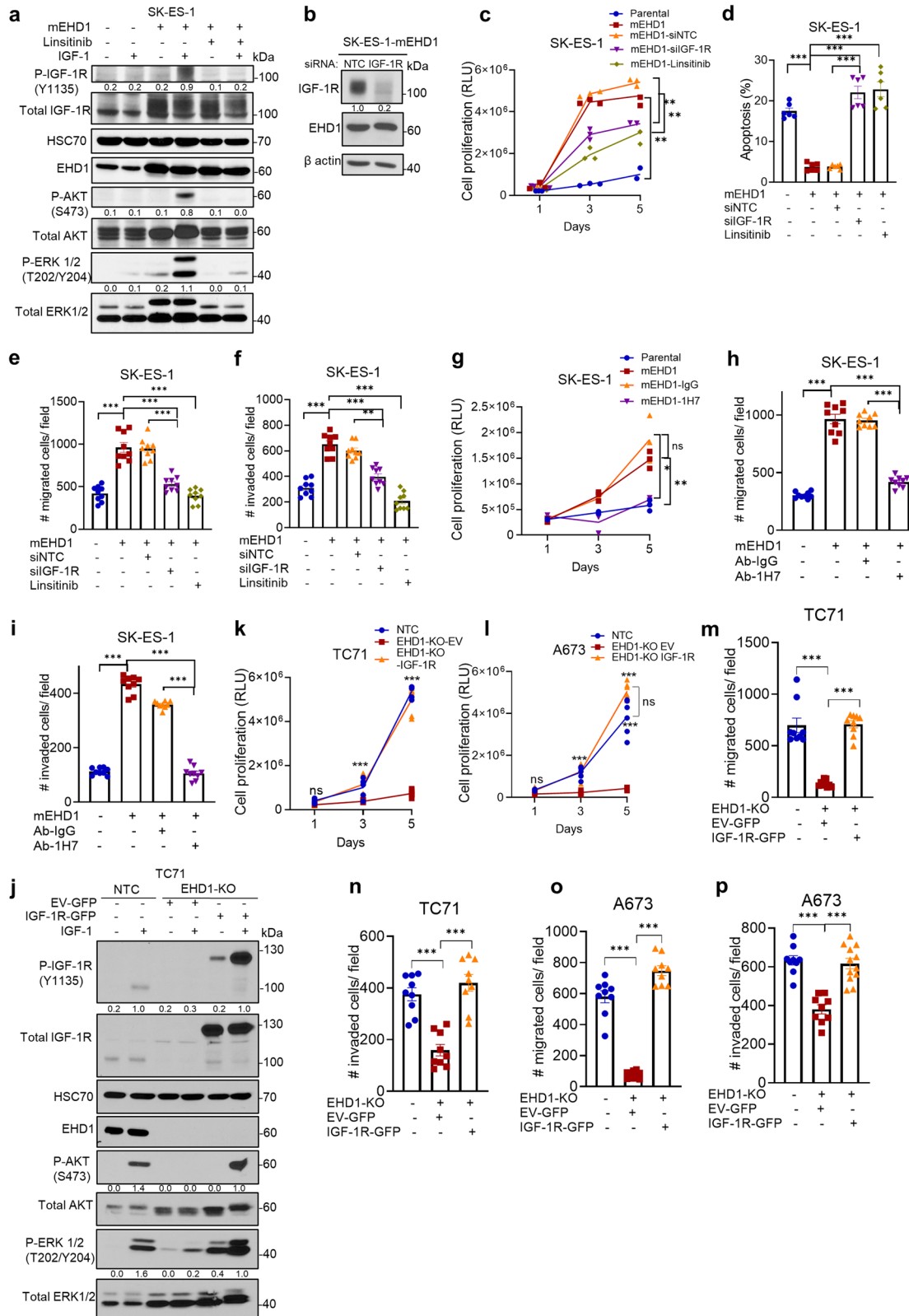

eurobonet.eu)]. All cases were genetically confirmed as belonging to the ESFT by molecular biology and/or fluorescent in situ hybridization (FISH). Approval for data acquisition and analysis was obtained from the Ethics Committees of the institutions involved in the study. All relevant ethical regulations were followed and informed consent was obtained. The clinical data were reviewed and stored within a specific database. Characteristics of the cohort and relevant clinical information have been previously reported[81]. A total of 24 tissue microarrays (TMAs) containing two representative cores for each case (1 mm in diameter)

were constructed for immunohistochemical analysis. Out of 324 samples, 307 and 227 samples could be analyzed for EHD1 and IGF-1R IHC expression, respectively. The deparaffinized sections were stained as per standard IHC protocol. Immunoreactivity was defined as follows: negative, fewer than 5% of tumor cells stained; poorly positive (score 1), between 5 and 10% of tumor cells stained; moderately positive (score 2), between 10 and 50% of tumor cells stained, and strongly positive (score 3), with more than 50% of the tumor cells were stained.

**Fig. 8 EHD1-dependent upregulation of oncogenic attributes of EWS cells requires the IGF-1R. a** Elevated IGF-1 signaling upon mEHD1 overexpression in EHD1-low SK-ES-1 cells requires IGF-1R expression and activity. Western blot analysis of phosphorylation of IGF-1R and key signaling pathway reporters (AKT and ERK1/2) in SK-ES-1-mEHD1 cell line. Cells were pre-starved for 48 h in serum/IGF-free medium combined with treatment with DMSO or 1 μM Linsitinib and either left unstimulated or stimulated with IGF-1 (50 ng/ml) for 30 min. **b–i** Mouse EHD1 (mEHD1)-overexpressing SK-ES-1 cell line was used to assess the requirement of IGF-1R in EHD1-driven pro-oncogenic attributes. SK-ES-1-mEHD1 cells transiently transfected with non-targeting control (NTC) or IGF-1R-targeted siRNA or treated with IGF-1R inhibitor linsitinib (1 μM) were studied for indicated traits. **b** Representative western blot confirming the effective IGF-1R knockdown upon transient IGF-1R siRNA relative to NTC siRNA transfection. **c** Elevated cell proliferation upon mEHD1 overexpression requires IGF-1R expression and activity. The SK-ES-1-mEHD1 cells were analyzed for IGF-1 (100 ng/ml)-dependent cell proliferation by Cell-Titer Glo assay with or without the indicated treatments. Parental SK-ES-1 cells without any treatments provided a baseline of cell proliferation without mEHD1 overexpression. **d** Elevated cell survival upon mEHD1 overexpression requires IGF-1R expression and activity. The SK-ES-1-mEHD1 cells grown in the presence of IGF-1 (100 ng/ml) without or with the indicated treatments for 3 days were analyzed for the proportion of apoptotic cells by FACS after Annexin-V and PI staining. **e, f** Elevated cell migration and invasion upon mEHD1 overexpression requires IGF-1R expression and activity. The SK-ES-1-mEHD1 cells were analyzed for IGF-1 (100 ng/ml)-dependent trans-well cell migration or invasion without or with the indicated treatments. Parental SK-ES-1 cells without any treatments provided a baseline of cell migration without mEHD1 overexpression. **g–i** Impact of treatment with IGF-1R inhibitory monoclonal antibody 1H7 (4 μg/ml) and its corresponding IgG control on cell proliferation (**g**), migration (**h**), and invasion (**i**) of mouse EHD1 (mEHD1)-overexpressing SK-ES-1 cell line. Mean +/− SEM of 3 experiments, each in triplicates. $*p < 0.05$; $**p < 0.01$; $***p < 0.001$; ns, not significant. **j** Rescue of signaling by exogenous IGF-1R expression in EHD1-KO EWS cells. EHD1-KO TC71 cells were transfected with Empty vector (EV)-GFP or IGF-1R-GFP constructs, and stable lines selected in G418 were pre-starved for 24 h in low serum/IGF-free medium and either left unstimulated or stimulated with IGF-1 (50 ng/ml) for 30 mins. Western blot analysis of rescue of phosphorylation of key signaling pathway reporters (AKT and ERK1/2) in IGF-1R-GFP as compared to NTC and EHD1-KO EV-GFP cell lines is shown. IGF-1 treatment also induced robust phosphorylation of introduced GFP-IGF-1R. **k–p** Rescue of cell proliferation (**k, l**), cell migration (**m, o**), and invasion (**n, p**) in TC71 and A673 EHD1-KO cells upon GFP-IGF-1R overexpression as compared to control EV-transfected cell lines. Data represent the mean +/− SEM of 3 independent experiments. $***p < 0.001$.

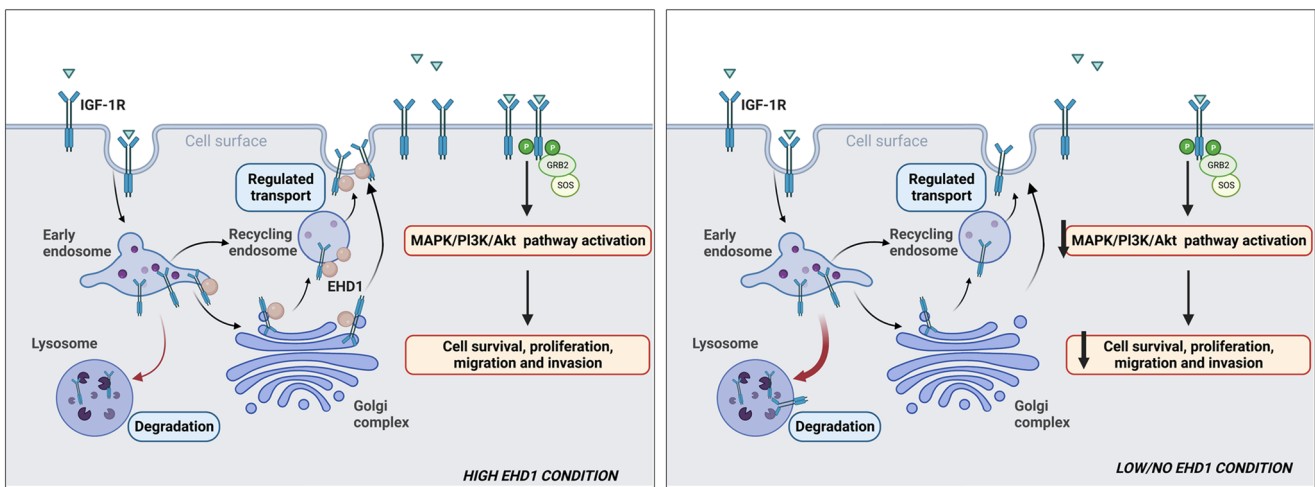

**Fig. 9 A model of how the EHD1/IGF-1R axis promotes IGF-1R-mediated signaling and tumor progression in Ewing Sarcoma.** EHD1 overexpression enhances the endocytic recycling and Golgi to plasma membrane transport of IGF-1R to elevate the cell surface receptor levels, thus enhancing IGF-1R-dependent signaling. Loss of EHD1 leads to IGF-1R mistargeting to lysosomes where it is degraded, resulting in reduced cell surface IGF-1R, diminished IGF-1R signaling and impaired tumorigenesis (Created with BioRender.com).

**Cell lines and medium**. Human Ewing Sarcoma cell lines TC-71, MHH-ES-1, and A4573 were obtained from Dr. Jason Yustein's laboratory at Baylor College of Medicine (TC-71, MHH-ES-1:DSMZ-German collection, A4573: Cellonco) and cultured in complete RPMI medium (Hyclone; #SH30027.02) with 10% fetal bovine serum (Gibco; #10437-028), 10 mM HEPES (Hyclone; #SH30237.01), 1 mM each of sodium pyruvate (Corning; #25-000-CI), nonessential amino acids (Hyclone; #SH30238.01) and L-glutamine (Gibco; #25030-081), 50 μM 2-ME (Gibco; #21985-023), and 1% penicillin/streptomycin (#15140-122; Gibco). A673 and SK-ES-1 cells were obtained from ATCC and cultured in complete DMEM medium (Gibco; #11965-092), and complete RPMI medium, respectively, supplemented as above. HEK-293T cells (ATCC CRL-3216) were cultured in complete DMEM medium. Cell lines were maintained for less than 30 days in continuous culture and were regularly tested for mycoplasma.

**Reagents and antibodies**. Primary antibodies used for immunoblotting were as follows: anti-HSC70 (#sc-7298) from Santa Cruz Biotechnology; anti-IGF-1Rβ (#3018), anti-IGF-1Rα (#17174), anti-phospho-IGF-1R-Y1135 (#3918), anti-phospho-AKT-S473 (#4060), anti-AKT (#4685), anti-ERK1/2 (#4695), anti-phospho-ERK1/2- Thr202/Tyr204 (#9101) from Cell Signaling Technology; and anti-beta-actin (#A5441) from Sigma. In-house generated Protein G-purified rabbit polyclonal anti-EHD1, EHD2, EHD3 and EHD4 antibodies have been described previously[2]. The horseradish peroxidase (HRP)-conjugated Protein A

(#101023) and HRP-conjugated rabbit anti-mouse secondary antibody (#31430) for immunoblotting were from Thermo Fisher. Antibodies used for immunofluorescence studies were as follows: anti-EHD1 (#ab109311) from Abcam; Alexa-555-conjugated anti-GM130 (#48641), anti-LAMP1 (#9091) and anti-RAB11 (#5589) from Cell Signaling Technology; anti-IGF-1Rβ (#MA5-13802) and Alexa-647-conjugated Wheat Germ Agglutinin (WGA) (#W32466) from Invitrogen. Secondary antibodies used for immunofluorescence studies were Alexa Fluor 594-conjugated goat anti-rabbit IgG (H + L) (#A11012) or Alexa Fluor 488-conjugated goat anti-mouse IgG (H + L) (#A11001) from Life Technologies Corporation. The Annexin-V-PI flow cytometric analysis was done using a kit (#V13241) from Invitrogen. Primary antibodies used for immunohistochemical studies included: anti-IGF-1R (#14534) and anti-cleaved-caspase 3 (#9661) from Cell Signaling Technology; and anti-CD99 (#ab-227738) and anti-Ki67 (#ab92353) from Abcam. For immunoprecipitation studies, primary antibodies included: anti-IGF-1Rβ (Cell Signaling Technology; #9750), anti-EHD1 (Abcam; #ab109311), and anti-Rabbit-IgG (Invitrogen; #02-6102). The sources for other reagents were as follows: PNGase F (NEB; #P0704S), Lambda Protein Phosphatase (NEB; #P0753S), cycloheximide (Sigma; #C7698); bafilomycin-A1 (SelleckChem; #S1413); linsitinib (SelleckChem; #S1091); recombinant-human-IGF-1 (Peprotech; #100-11); IGF-1 Receptor α mAb(1H7) (Santa Cruz; #sc-461); doxycycline (Sigma Aldrich; #D9891); aprotinin (Sigma Aldrich; #A1153); and leupeptin (Sigma Aldrich; #L2884).

**Generation of knockdown, CRISPR knockout, and luciferase reporter cell lines.** To generate stable doxycycline-inducible EHD1-shRNA and non-targeting control (NTC)-shRNA expressing TC71, A673, and SK-ES-1 cell lines, the following lentiviral SMART-vector constructs encoding a GFP and human EHD1-shRNA (#V3SH11252-229594140, #V3SH11252-225446205 and #V3SH11252-228109140, designated shEHD1 #1, #2, and #3, respectively) or an NTC-shRNA were obtained from Dharmacon. Lentiviral supernatants were generated by transient co-transfection of individual constructs with packaging plasmids (psPAX2, Addgene #12260 and pMD2.G, Addgene #12259 into HEK-293T cells using X-tremeGENE HP DNA transfection reagent (#06366236001; Roche). The supernatants were applied to cells for 48 h in the presence of polybrene (10 μg/ml, Sigma #H9268) and stable polyclonal cell lines were selected with 1 μg/ml puromycin and maintained in their respective media with tetracycline-free 10% FBS (Novus Biologicals #S10350) and 1 μg/ml puromycin. For CRISPR-Cas9 mediated gene editing, the EHD1 sgRNA CRISPR/Cas9 All-in-One Lentivector (pLenti-U6-sgRNA-SFFV-Cas9-2A-Puro; #K0663105) or Scrambled sgRNA CRISPR/Cas9 All-in-One Lentivector (#K010) from Applied Biological Materials were used to generate lentiviral supernatants that were transduced into TC71 or A673 cell lines followed by selection with 1 μg/ml puromycin. Clonal derivatives were obtained by limiting dilution and screened for complete knockout using western blotting. Unless otherwise indicated, 3 or 4 clones (maintained separately) representing two EHD1 sgRNA targets were pooled for experimental analyses. For rescue experiments, the mouse *Ehd1* lentiviral vector (pLenti-GIII-CMV-RFP-2A-Puro) (#190510640495; Applied Biological Materials) was stably transduced into TC71-EHD1-KO, A673-EHD1-KO and SK-ES-1 cell lines followed by selection with 1 μg/ml puromycin. The tdTomato-luciferase plasmid was generated by recombineering using the following pMuLE system plasmids from Addgene: pMuLE ENTR U6-miR-30 L1-R5 (#62113); pMuLE ENTR SV40 tdTomato L5-L2 (#62157) and pMuLE Lenti Dest Luc2 (#62179). The mCherry-luciferase plasmid (pCDH-EF-eFFly-T2A-mCherry; Addgene #104833) was used to generate lentiviral supernatants that were transduced into the indicated cell lines followed by FACS sorting of mCherry-high fraction. *EHD1* knockout sites were assessed by Sanger sequencing of PCR fragments generated with genomic DNA as template with the following primers: 5′-AGTGTGGGTCGCTCCCG-3′ (forward) and 3′-GAGGAGCACCATAGGCTTGT-5′ (reverse). For IGF-1R overexpression in EHD1-KO cell lines, a Turbo-GFP-tagged human *IGF1R* cDNA construct (#RG214928; Origene) or pCMV6-AC-GFP empty vector (#PS100010; Origene) were transfected into cells followed by selection of polyclonal lines in 300 μg/ml Neomycin G418 (#A1720; Sigma). For IGF-1R siRNA knockdown, ON-TARGETplus SMARTpool siRNA (#L-003012-00-0005), ON-TARGETplus Non-targeting pool(#D-001810-10-05) were transiently transfected into cells using Dharmafect I transfection reagent (#T-2001-01) (all from Dharmacon – Horizon Discovery).

**Western blotting.** Whole-cell extracts were prepared, and western blot was performed as described previously[5] with minor modifications. Cells were lysed in Triton-X-100 lysis buffer (50 mM Tris pH 7.5, 150 mM NaCl, 0.5% Triton-X-100, 1 mM PMSF, 10 mM NaF, 1 mM sodium orthovanadate, 10 μg/ml each of Aprotinin and Leupeptin) Lysates were rocked at 4 °C for >1 h, spun at 13,000 rpm for 30 min at 4 °C and supernatant protein concentration determined using the BCA assay kit (#23225; Thermo Fisher Scientific). 30–50 μg aliquots of lysate proteins were resolved on sodium dodecyl sulfate-7.5% polyacrylamide gel electrophoresis (SDS-PAGE), transferred to polyvinylidene fluoride (PVDF) membrane, and immunoblotted with the indicated antibodies.

**Immunoprecipitation (IP).** 500 μg-1 mg aliquots of cleared lysate protein were incubated with optimized amounts of the indicated antibodies and rocked overnight at 4 °C. 60 μl of PBS-pre-washed and PBS/1% BSA blocked protein A-Sepharose beads (#101042; Invitrogen) were added to each sample and rocked overnight at 4 °C. The beads were washed six times with TX-100 lysis buffer, and bound proteins were resolved by SDS–7.5% PAGE, transferred to PVDF membrane, and immunoblotted with indicated primary antibodies. 50 μg aliquots of whole-cell lysates were run as input controls.

**Live-cell surface biotin labeling to assess the cell surface IGF-1R levels.** Cell monolayers were washed with ice-cold PBS, and incubated in PBS with 2 mM EZ-Link Sulfo-NHS-LC-Biotin (#A39257; Thermo Fisher) for 30 min at 4 °C. The cells were washed in PBS and their TX-100 lysates subjected to anti-IGF-1R or control IgG immunoprecipitation followed by blotting with Streptavidin-Horseradish Peroxidase (HRP) Conjugate (# SA10001; Thermo Fisher) and chemiluminescence detection.

**Immunofluorescence.** Cells plated on Poly-L-lysine coated coverslips were treated as indicated in figure legends, fixed using 4% paraformaldehyde in PBS for 20 min at RT. Cells were then permeabilized in 0.3% Triton X-100 for 20 min at room temperature, blocked with 10% goat serum in PBS, and incubated with primary antibodies in 1% goat serum and 1% BSA in PBS at 4 °C overnight. After washing in 0.1% BSA-PBS, the cells were incubated with the appropriate fluorochrome-conjugated secondary antibody for 1 h at RT, washed 0.1% BSA-PBS and mounted

using Vectashield-mounting medium with DAPI (Vector Laboratories; #H-1500). For wheat-germ agglutinin staining, after fixation, cells were washed with PBS and incubated with Alexa Fluor® 647-WGA conjugate concentration of 5.0 μg/mL for 10 min at room temperature. Cells were washed with PBS followed by permeabilization for subsequent counterstaining. Confocal images were captured using a Zeiss LSM 800 with microscope Airyscan. Merged pictures were generated using ZEN 2012 software from Carl Zeiss and fluorescence intensities were quantified using the ImageJ (NIH) software. The % IGF-1R transport to the cell surface was calculated as a ratio of the mean fluorescence intensity at the cell surface (using a freeform selection tool) to the mean fluorescence intensity of the entire cell. The background mean fluorescence was measured by selecting a region next to the cell of interest that showed no fluorescence and this value was subtracted from the cell fluorescence readings. Pearson's correlation coefficients of co-localization were analyzed using the ImageJ JACoP colocalization analysis module. A threshold was established first using the JACoP threshold optimizer, followed by calculation of Pearson's correlation coefficients.

**Quantification of cell surface IGF-1R using FACS analysis.** $2 \times 10^5$ cells were seeded per well of six-well plates and grown in regular medium with 10% FBS for 48 h. Cells were further treated as indicated in figure legends, rinsed with ice-cold PBS, released from dishes with trypsin-EDTA (#15400054; LifeTech (Thermo-Fisher) and the trypsinization stopped by adding equal volume of soybean trypsin inhibitor (#17075029; LifeTech (ThermoFisher). Cells were washed thrice in ice-cold FACS buffer (1% BSA in PBS), and live cells stained with PE-anti-human-IGF-1R (#351806; Biolegend) or PE-Mouse-IgG isotype control (#400112; Biolegend). FACS analyses were performed on a LSRFortessa X50 instrument and data analyzed using the FlowJo software. The gating strategy has been included in Supplementary Fig. S14.

**Transwell migration and invasion assay.** For migration and invasion assays, $2 \times 10^5$ cells were seeded in top chambers of regular or Matrigel-coated transwells (migration – Corning #353097; invasion – Corning #354480) in 400 μl of 0.5% FBS-containing medium for 3 h before migration/invasion towards medium containing 10% FBS or 100 ng/ml IGF-1 in lower chambers, as indicated in figure legends. Both the top and lower chamber media contained Mitomycin C (10 μg/ml) to eliminate the contribution of cell proliferation. After 16 h, the cells on the upper surface of the membranes were scraped with cotton swabs, and the migrated cells on the bottom surface were fixed and stained in 0.5% crystal violet in methanol. Five randomly selected visual fields on each insert were photographed, and cells were enumerated using the ImageJ software. Each experiment was run in triplicates and repeated three times.

**Cell proliferation assay.** 500 cells/well were seeded in 96-well flat-bottom plates in 100 ml medium and an equal volume of the CellTiter-Glo Luminescent Assay Reagent (#G7571; Promega) added at the indicated time points. Luminescence was recorded using a GloMax® luminometer (Promega).

**Anchorage independent growth assay.** $10^4$ cells suspended in 0.4% soft agar were plated on top of a pre-solidified 0.8% soft agar bottom layer in 6-well plates. After two weeks, cells were fixed and stained with 0.5% crystal violet in methanol and imaged under a phase contrast microscope. The number of colonies in the entire well were quantified using the ImageJ software. All experiments were done in triplicates and repeated three times.

**Tumorsphere assay.** Cells were suspended in DMEM/F12 media (Thermo Fisher; #1133032) supplemented with 1% penicillin/streptomycin, 4 μg/ml heparin (Stem cell technologies; #07980), 20 ng/ml Animal-Free Recombinant Human EGF (Peprotech; #AF-100-15), 10 ng/ml Recombinant Human FGF-basic (Peprotech; #100-18B), 1X N-2 supplement (Gibco; #17502-048), 1X B27 supplement (Gibco; #17504-044) and 4% Matrigel (BD Biosciences; #356234) and seeded at $10^4$/well in ultra-low attachment 24-well plates. After one week, tumorspheres were imaged under a phase contrast microscope. Tumor-spheres greater than 40 μm in diameter were quantified using the ImageJ software. All experiments were done in triplicates and repeated 3 times.

**RNA sequencing and enrichment analysis of differentially expressed genes.** Total RNA was isolated using Qiagen RNeasy RNA extraction kit (#74104) and further cleaned using the RNeasy PowerClean Pro Cleanup kit (#13997-50), as per manufacturer's protocols. The purity of RNA was assessed on a Bioanalyzer in the UNMC Next Generation Sequencing Facility. 1 μg of cleaned RNA samples were used to generate RNA-seq libraries using the TruSeq RNA Library Prep Kit v2 (Illumina) following the manufacturer's protocols and sequenced using the $2 \times 75$ bases paired-end protocol on a NextSeq550 instrument (Illumina). For differential expression analysis, paired-end reads were aligned to the human genome version hg38 using hisat2 guided by Ensembl gene annotations[82] and annotated transcripts were quantified and TPM normalized using Stringtie 2.1.1[83] Differential expression was assessed by DESeq2[84] and significantly changed genes were required to have a Benjamini–Hochberg adjusted p-value of <0.05 and a 2-fold change in expression.

Gene Set Enrichment Analysis (GSEA) and pathway analyses were performed using MSigDB and Ingenuity-Pathway Analysis (IPA).

**RNA isolation and real-time PCR analysis**. Total RNA was extracted from cells using the Qiagen RNeasy RNA extraction kit (#74104) as per manufacturer's protocols. cDNA was obtained by reverse transcription using the QuantiTect Reverse Transcription kit (Qiagen; #205311) and real-time qPCR was performed using the SYBR Green labeling method (Qiagen; QuantiTect SYBR Green PCR kit #204143) on an Applied Bioscience QuantStudio thermocycler. The primer sequences (Integrated DNA Technologies) for qRT-PCR were: human *IGF1R* 5′-TCTGGCTTGATTGGTCTGGC-3′ (forward), 5′-AACCATTGGCTGTG-CAGTCA-3′ (reverse); *PCNA* 5′-AGCAGAGTGGTCGTTGTCTTT-3′ (forward), 5′-TAGGTGTCGAAGCCCTCAGA-3′ (reverse); *E2F1* 5′-CGCCATCCAG-GAAAAGGTGT-3′ (forward), 5′-AAGCGCTTGGTGGTCAGATT-3′ (reverse); *E2F2* 5′-CAACATCCAGTGGGTAGGCA-3′ (forward), 5′-TGCTCCGTGTTCATCAGCTC-3′ (reverse); *CDK4* 5′-TGTATGGGGCCGTAG-GAAC-3′ (forward), 5′-TCCAGTCGCCTCAGTAAAGC-3′ (reverse); *CDK6* 5′-ACCCACAGAAACCATAAAGGATA-3′ (forward), 5′-GCGGTTTCA-GATCACGATGC-3′ (reverse). The fold change of gene expression was calculated relative to the control using the ΔΔCt method and normalized to *GAPDH*.

**Phospho-RTK array analysis**. The Human Phospho-RTK Array Kit from R&D systems (#ARY001B) was used. Cells grown to 80% confluency were lysed and 300 μg of lysate protein were applied to supplied arrays and processed according to manufacturer's instructions. Signals corresponding to 49 tyrosine phosphorylated RTKs on the array were visualized using chemiluminescence and analyzed using ImageJ software; average signal (pixel density) of duplicate spots was used to calculate fold differences.

**Xenograft studies and IVIS imaging**. All animal experiments were performed with the approval of the UNMC Institutional Animal Care and Use Committee (IACUC Protocol 19-017-04-FC). For analyses of EHD1-knockdown cell implants, 6-week-old female athymic nude mice (Charles River) were injected via the intratibial route with $10^6$ cells (in 100 μl cold PBS) engineered with lentiviral tdTomato-luciferase. Once palpable tumors were observed, the mice were randomly assigned into minus (−) Dox or plus (+) Dox groups (Dox at 2 mg/ml in drinking water with 1% sucrose). For analyses of EHD1-KO and mEHD1-rescued cell implants, 6-week-old male athymic nude mice were injected via the intratibial route with $2 \times 10^5$ cells (in 20 μl cold PBS) engineered with lentiviral mCherry-enhanced luciferase. Tumor growth was monitored biweekly for up to 30 days using calipers, with tumor volume calculated from length × width$^2$/2. For bioluminescent imaging, mice received an intraperitoneal injection of 200 μl D-luciferin (15 mg/ml; Millipore Sigma #L9504) 15 min before isoflurane anesthesia and were placed dorso-ventrally in the IVIS™ Imaging System (IVIS 2000). Images were acquired using the IVIS Spectrum CT and analyzed using the Living Image 4.4 software (PerkinElmer). Mice were imaged weekly and followed for up to 30 days. At the end of the study, mice were euthanized, and hind limbs, lungs, and livers were harvested. Bioluminescent signals from the harvested lungs and livers were recorded for analyses of tumor metastasis. Resected tumor xenografts were fixed in formalin, and paraffin-embedded tissue sections were used to perform the immunohistochemical staining.

**Bone morphometry analysis by micro-CT**. The hind legs of mice harvested post-euthanasia were fixed in formalin and scanned using a micro-CT instrument (Skyscan 1172, Bruker). The parameters were 55 kV, 181 μA, 0.5 mm aluminum filter, 9 μm resolution, 4 frames averaging, 0.4 rotation step, 180° scanning. The raw images were reconstructed using the NRecon software (version 1.7.4.6, Bruker microCT). All reconstructed images were registered and realigned before analysis using the DataViewer software (version 1.5.6.2, Bruker microCT). The tibial bone was then evaluated using CTAn software (version 1.18.8.0, Bruker microCT) to calculate the percent bone volume (BV/TV), trabecular thickness (Tb.Th), trabecular number (Tb.N), and trabecular separation (Tb.Sp).

**Statistics and reproducibility**. GraphPad Prism software (version 8.0.2) was employed to perform all the statistical analyses. Statistical analyses of in vitro data were performed by comparing two groups using two-tailed student's t test. A two-way ANOVA test was used to analyze the in vivo mouse tumor growth. $P$ values equal to or <0.05 were considered significant. For patient tissue sample analyses, association with categorical histopathological parameters was assessed using a chi-square test to determine homogeneity or linear trend for ordinal variables. The significance level was set at 5%. To study the impact of the histological, immunohistochemical, and molecular factors on progression-free survival (PFS) and disease-specific survival (DSS), the Kaplan–Meier proportional risk test (log rank) was used.

**Reporting summary**. Further information on research design is available in the Nature Portfolio Reporting Summary linked to this article.

## Data availability

RNA-seq data have been deposited at the Gene Expression Omnibus (GEO; GSE225214). Numerical source data for graphs and charts has been uploaded to the file Supplementary Data. Supplementary Figure S13 contains all the original western blots.

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

## Acknowledgements

This research was funded by: Pilot grants to H.B. from the Pediatric Cancer Research Program of the Children's Hospital Research Institute, University of Nebraska Medical Center; Pilot grants from the Fred & Pamela Buffett Cancer Center (H.B. & V.B.); Department of Defense grants W81XWH-17-1-0616 and W81XWH-20-1-0058 to H.B. and W81XWH-20-1-0546 to V.B.; the NIH grants R21CA241055 and R03CA253193 to V.B.; NIH/NIGMS grant R35GM147467 MIRA to M.J.R.; the Raphael Bonita Memorial Fund to H.B.; and support to UNMC core facilities from the NCI Cancer Center Support Grant (P30CA036727) awarded to Fred & Pamela Buffett Cancer Center and from the Nebraska Research Initiative. S.C. and A.M.B. received the University of Nebraska Medical Center Graduate Student Fellowships.

## Author contributions

Designing research studies: S.C., H.B., V.B., and B.C.M. Conducting experiments: S.C., A.M.B., I.M., H.L., and B.C.M. Acquiring and analyzing data: S.C., B.C.M., A.K., S.M., N.C., J.A.L., A.L.B., I.M., J.L.M., D.W.C., and M.J.R. Providing reagents: M.D.S., K.S., I.M., and G.G. Writing the manuscript: S.C. and H.B. All authors have read and agreed with this version of the manuscript.

## Competing interests

Dr. H. Band and Dr. V. Band received funding from Nimbus Therapeutics for an unrelated project. The authors declare no competing interests.
