## [Peer Review File · Communications Biology]

Reviewers' comments:

Reviewer #1 (Remarks to the Author):

To authors

The present manuscript by Chakraborty et. al deals with the effects of EHD1 on Ewing sarcoma tumorigenesis and metastasis. These effects were assessed by several methods including cell proliferation/apoptosis/migration assays, western blots, fluorescent microscopy analysis, RNA-seq and animal experiments. Authors concluded that EHD1 overexpression is associated with poor prognosis in Ewing sarcoma and proper transport of IGF1R to the cell surface by EHD1 is required for EHD1 overexpression-dependent oncogenesis. The model proposed in Fig 8J can be reasonably inferred from the data but needs a little more mechanistic work for the paper to be accepted.

My comment is follows:

1. In Fig. 4C, EHD1 knockdown decreased IGF1R, whereas rescue of EHD1 (overexpression of EHD1) increased IGF1R. IGF1R bands in both rescue cells were clearly shifted up compared with their non-treated cells, suggesting EHD1 overexpression cause post-translational modifications (ubiquitination/SUMOylation, and glycosylation) and may modify intracellular localization and downstream signaling of IGF1R. The authors could address whether EHD1 overexpression affect modifications, intracellular localization and signaling of IGF1R.

2. In Fig. 4D, the authors quantified IGF1R on the cell surface by FACS analysis. According to the materials and methods (line 207-214), the authors used trypsin-EDTA to dissociate cells. Trypsin can digest extracellular domains of cell-surface proteins including IGF1R. PE-anti-human IGF1R antibody that you used for FACS analysis recognizes a fnIII domain in the alpha subunit of IGF1R. There is a possibility that it may be digested by trypsin. Therefore, I am wondering whether obtained signals came from only cell surface IGF1R. The authors should address whether alpha subunit of IGF1R was not degraded by trypsin-treatment using western blot.

Reviewer #2 (Remarks to the Author):

S. Chakraborty and colleagues have demonstrated in the present study that a high expression levels of EHD1 in EWS was closely linked to its malignancy, using in vitro genetic engineering and in vivo tumor transplanting experiments. The high expression of EHD1 in EWS cells could change the traffic itinerary of newly synthesized IGF-1R, targeting to Golgi-to-plasma-membrane path, and, consequently, sustained IGF-1R in the cell surface caused an overactivation of downstream signals, which brought cells a metastatic property.

The experiments are well designed, and the results and conclusions are very clear. The manuscript is written logically and having good readability. However, more experiments are needed to show that IGF receptor traffic regulation by EHD1 is a causative factor in tumorigenesis or metastasis.

Major and minor comments are below:

Major comments

1, Authors clearly showed that EHD1 is essential for tumorigenesis and that EHD1 plays important roles in traffic of IGF-I receptor to the cell surface. However, there is no reliable evidence whether increased IGF-I signaling is responsible for the tumorigenesis caused by elevated EHD1 expression. To prove this, it must be shown that EHD1 KO can be restored or the effects of EHD1 rescue cannot be seen by changing IGF-I receptor traffic in other ways, rather than by dropping IGF signaling (IGF-I

receptor inhibitor). Even if that is difficult, authors need to measure IGF signaling activation (e.g. pAkt, pErk) and traffics not only in EHD1KD (or KO) cells (Fig 7A,B) but also in EHD1 rescue cells (EHD1 overexpression cells) and in IGF-I receptor inhibitor treated cells, and examine their correlation and discuss thoroughly. Specifically, it should be examined whether IGF signaling is enhanced in EHD1 rescue cells. Furthermore, which has lower IGF signaling in IGF receptor inhibitor-treated cells or EHD1KD cells.

2, The impact of reduced cell proliferation on tumorigenesis and malignancy should also be well discussed.

At first, I do not see any data mentioned for Lane 344-347. This data should be clearly indicated. Furthermore, EHD1 KO and IGF-I receptor inhibitors hardly activate IGF signaling. Wouldn't it be obvious that tumorigenesis would not be observed under the conditions which IGF signal is completely inhibited? For even normal cells it is difficult to proliferate or survive if IGF signal is blocked. I would like to see thorough discussion of this issue.

3, An EHD1-dependent mis-traffic of IGF-1R to the cell surface and sustaining it in the cell surface is a key element in the authors' scenario. But data illustrating the existence of IGF-1R in the cell surface seems to be relatively small in Figure 4B. I recommend a membrane fractionation experiment or surface biotinylation assay to strengthen your conclusion.

4, Fig 1D indicates that ~90% of clinical tissue samples had a high EHD1 expression, while only ~60% of them show high IGF-1R expressions (Fig 4G). Please provide an explanation regarding this gap.

5, The authors mention that IGF-1R targeting clinical trials have been explored but the results have been disappointing. However, EHD1-KD/KO (presumably equivalent to IGF-IR inhibition) diminished tumor growth in transplanted mice, and pharmacological inhibition of IGF-IR indeed significantly decreased tumor malignancy (Fig 3 and 8). Please give the authors' opinion as to why IGF-IR targeting strategy has not been successful and therapeutic application of EHD1-dependent mechanism.

6, In Figure 5B, how did the authors measure the ratio of IGF receptor recycling to the plasma membrane? Traffic to the recycling endosome is measured by co-localization with Rab11. The localization of the receptor to the plasma membrane should likewise be shown by co-localization with other co-staining.

And the immunostaining signal (Figure 5, 6, S5, S6) is too strong. So, what should appear as dots appear to be squashed. The signal should be weakened to clearly show the localization.

Minor comments

4, Please provide more information of CCLE (Line#326).

5, Is what the sentence (Line#592-594) mention Fig 8? Is it correct?

6, Some abbreviations used in the manuscript appear without any explanations (e.g. ESFT, Line#103). Please read through again.

Reviewer #3 (Remarks to the Author):

The paper by Chakraborty et al. investigates the role of EHD1 (EPS15 Homology Domain containing 1) protein in oncogenesis. The authors showed increased expression of EHD1 in patient samples, both primary tumors and metastasis. Use of shRNA and CRISPR mediated downregulation showed EHD1 role in proliferation, migration, invasion in vitro and tumor growth and metastasis in vivo. Mechanistically, the authors showed that EHD1 regulates surface localization of IGF1R (IGF1 receptor)

and activation. The study is very well conducted and their results on co-targeting these two pathways is very translational. There was no major concern noted in the presented paper.

Minor concerns:

The authors should explain the shift in band of IGF-1R in Fig 4C. Is the phosphorylation mediated shift.

The authors need to describe figure 8 in more detail (each panel). It is very hard to read.

Response to Reviewer Comments:

The addition of new data has resulted in renumbering as indicated below (new figures are also indicated). New text in the main article file has also been indicated in “blue”.

Previous figure number	New Figure number	New data
Figure 4d	Figure 4f-h	Figure 4d-e
Figure 4e	Figure 4i	Figure 4j
Figure 4f	Figure 4k	Figure 7c-f
Figure 4g	Figure 4l	Figure 8a
Figure 4h	Figure 4m	Figure 8j-p
Figure 4i	Figure 4n	Supplementary Figure S2d
Figure 7e	Figure 7g	Supplementary Figure S7b-e
Figure 7f	Figure 7j-k	Supplementary Figure S10a-e
Figure 7g	Figure 7l	Supplementary Figure S11a-d
Figure 7h	Figure 7h-i	Supplementary Figure S12h-i
Figure 8a	Figure 8b	
Figure 8j	Figure 9	
Supplementary Figure S1a	Supplementary Figure S1	
Supplementary Figure S1b	Supplementary Figure S2a	
Supplementary Figure S1c-d	Supplementary Figure S2b-c	
Supplementary Figure S1k	Supplementary Figure S3	
Supplementary Figure S2	Supplementary Figure S4	
Supplementary Figure S3	Supplementary Figure S5	
Supplementary Figure S4	Supplementary Figure S6	
Supplementary Figure S5a	Supplementary Figure S7a	
Supplementary Figure S5b	Supplementary Figure S5f	
Supplementary Figure S6	Supplementary Figure S8	
Supplementary Figure S7	Supplementary Figure S9	
Supplementary Figure S8a	Figure 7h	
Supplementary Figure S8b	Supplementary Figure S12a	
Supplementary Figure S8c	Supplementary Figure S12b	
Supplementary Figure S8d	Supplementary Figure S12c	
Supplementary Figure S8e	Supplementary Figure S12d-e	
Supplementary Figure S8f-g	Supplementary Figure S12f-g	

Reviewer #1 General Comments:

The present manuscript by Chakraborty et. al deals with the effects of EHD1 on Ewing sarcoma tumorigenesis and metastasis. These effects were assessed by several methods including cell proliferation/apoptosis/migration assays, western blots, fluorescent microscopy analysis, RNA-seq and animal experiments. Authors concluded that EHD1 overexpression is associated with poor prognosis in Ewing sarcoma and proper transport of IGF1R to the cell surface by EHD1 is required for EHD1 overexpression-dependent oncogenesis. The model proposed in Fig 8J can be reasonably inferred from the data but needs a little more mechanistic work for the paper to be accepted.

Reviewer #1 Specific Comments:

Comment #1. In Fig. 4C, EHD1 knockdown decreased IGF1R, whereas rescue of EHD1 (overexpression of EHD1) increased IGF1R. IGF1R bands in both rescue cells were clearly shifted up compared with their non-treated cells, suggesting EHD1 overexpression cause post-translational modifications (ubiquitination/SUMOylation, and glycosylation) and may modify intracellular localization and downstream signaling of IGF1R. The authors could address whether EHD1 overexpression affects modifications, intracellular localization and signaling of IGF1R.

Response to Comment #1: We thank the reviewer for their observation and suggestions. Our further survey of the literature strongly suggested the possibility that the mobility shift in IGF-1R upon ectopic EHD1 expression may reflect differential N-linked glycosylation. Absence of N-linked glycosylation of IGF-1R was shown to impair the membrane localization of IGF-1R and led to anti-IGF-1R antibody (figitumumab) insensitivity in gastric and hepatocellular cancer cell lines¹. Studies in Ewing Sarcoma cell lines have shown that inhibition of N-linked glycosylation of IGF-1R downregulated the plasma membrane bound IGF-1R and consequently decreased the IGF-1R signaling and EWS cell survival². The fully N-linked glycosylated band of IGF-1R in these publications resembled the upper IGF-1R species we see. To directly test this possibility, we treated the cells with PNGase F for enzymatic removal of N-linked oligosaccharides. In each cell line (TC71, A673 and SKES1), the up-shifted IGF-1R band in the "Rescue/Overexpression" lines shifted down and co-migrated with the predominant band seen in the parental cell lines (**new Figure 4d-e, new Supplementary Fig. S7b**). Thus, we conclude that the change in the mobility of IGF-1R upon EHD1 overexpression is due to its more complete N-linked glycosylation. We also explored a potential phosphorylation-mediated mobility shift in IGF-1R in the EHD1-rescue/overexpression cell lines. However, treatment with Lambda-phosphatase was unable to change the mobility of up-shifted IGF-1R bands (**new Supplementary Figure S7c-d**).

As suggested by the reviewer, we re-examined the impact on signaling including the EHD1 rescue/overexpression cell line versions. Indeed, EHD1 overexpression in the EHD1-low SK-ES-1 cell line enhanced the IGF-1 dependent MAPK and AKT phosphorylation (**new Figure 8a**). We also observed the EHD1 overexpression-dependent rescue of signaling defects in EHD1-KO TC71 and A673 cell lines (**new Figure 7c-d**).

Our prior studies of CSF-1R and EGFR, and the present manuscript on IGF-1R, support a key role of EHD1 in post-Golgi sorting of newly synthesized RTKs towards the plasma membrane, with loss of EHD1 promoting their lysosomal delivery and subsequent degradation. In light of the new results based on the reviewer-suggested experiments, and previous studies linking IGF-1R glycosylation to its plasma

membrane traffic, we suggest (**in the Discussion line# 712-726**) that EHD1's role in post-Golgi sorting of IGF-1R (and potentially other RTKs) may reflect more complete N-linked glycosylation. However, further studies are needed to test this notion.

Comment #2. In Fig. 4D, the authors quantified IGF1R on the cell surface by FACS analysis.

According to the materials and methods (line 207-214), the authors used trypsin-EDTA to dissociate cells. Trypsin can digest extracellular domains of cell-surface proteins including IGF1R. PE-anti-human IGF1R antibody that you used for FACS analysis recognizes a FnIII domain in the alpha subunit of IGF1R. There is a possibility that it may be digested by trypsin. Therefore, I am wondering whether obtained signals came from only cell surface IGF1R.

The authors should address whether alpha subunit of IGF1R was not degraded by trypsin-treatment using western blot.

Response to Comment #2: In the presented studies, live cell surface FACS analysis of IGF-1R involved a brief treatment with 0.05% trypsin-EDTA (as the cell line come off the plates quickly) and the trypsin was immediately neutralized with soybean trypsin inhibitor. As an internal control, cells treated with IGF-1, which promotes the ligand-induced internalization and degradation of IGF-1R (continued-IGF1 condition), showed an expected decrease in the cell surface IGF-1R staining compared to cells maintained under steady state or post-starvation only, confirming that the FACS signals measured represent the cell surface IGF-1R. Additionally, Western blot analysis using an alpha subunit specific antibody against IGF1R, as suggested by the reviewer, showed that trypsin/EDTA treatment used to dissociate cells did not lead to degradation of the IGF-1R alpha subunit (**new Supplementary Figure S7e**).

Reviewer #2 General Comments:

S. Chakraborty and colleagues have demonstrated in the present study that a high expression levels of EHD1 in EWS was closely linked to its malignancy, using in vitro genetic engineering and in vivo tumor transplanting experiments. The high expression of EHD1 in EWS cells could change the traffic itinerary of newly synthesized IGF-1R, targeting to Golgi-to-plasma-membrane path, and, consequently, sustained IGF-1R in the cell surface caused an overactivation of downstream signals, which brought cells a metastatic property.

The experiments are well designed, and the results and conclusions are very clear. The manuscript is written logically and having good readability. However, more experiments are needed to show that IGF receptor traffic regulation by EHD1 is a causative factor in tumorigenesis or metastasis.

Major and minor comments are below:

Reviewer #2 Major comments:

Comment #1. Authors clearly showed that EHD1 is essential for tumorigenesis and that EHD1 plays important roles in traffic of IGF-I receptor to the cell surface. However, there is no reliable evidence whether increased IGF-I signaling is responsible for the tumorigenesis caused by elevated EHD1 expression. To prove this, it must be shown that EHD1 KO can be restored or the effects of EHD1 rescue cannot be seen by changing IGF-I receptor traffic in other ways, rather than by dropping IGF signaling (IGF-I receptor inhibitor). Even if that is difficult, authors need to measure IGF signaling activation (e.g. pAkt, pErk) and traffics not only in EHD1KD (or KO) cells (Fig 7A,B) but also in EHD1 rescue cells (EHD1 overexpression cells) and in IGF-I receptor

inhibitor treated cells, and examine their correlation and discuss thoroughly. Specifically, it should be examined whether IGF signaling is enhanced in EHD1 rescue cells. Furthermore, which has lower IGF signaling in IGF receptor inhibitor-treated cells or EHD1KD cells.

Response to Comment #1: We thank the reviewer for their observation and suggestions. As suggested by the reviewer, we performed additional experiments to confirm that increased IGF-1R signaling mediates the effects of elevated EHD1 expression. We now show that reduced IGF-1-induced signaling in EHD1-KO cells is restored by exogenous IGF-1R-GFP expression, as seen by rescue of p-IGF-1R, p-AKT and p-ERK induction as well as cell proliferation, migration, and invasion responses to IGF-1 (**new Figure 8j-p, new Supplementary Figure S12h-i**). Conversely, EHD1 overexpression in EHD1-low SK-ES-1 cell lines enhanced the IGF-1-induced signaling, and the signaling defects seen in EHD1-KO TC71 and A673 cell lines were rescued in mEHD1-expressing versions of these cell lines (**new Figure 7c-d, new Figure 8a**). We further show that the signal enhancement caused by mEHD1 overexpression is brought down to control cell line levels by treatment with the IGF-1R inhibitor linsitinib (**new Figure 8a**). Combined with our findings that IGF-1R inhibition using siRNA knockdown of IGF-1R expression, kinase inhibition with linsitinib as well as treatment with an extracellular domain-specific inhibitory antibody 1H7 restores mEHD1 overexpression-mediated oncogenic phenotype including cell migration, invasion, proliferation, and survival in SK-ES-1 cells (**Figure 8b-i, Supplementary Figure S12a-g**), these new data further support a causal role of IGF-1R in EHD1 overexpression-induced oncogenic phenotype of EWS cells. Furthermore, our results that the EHD1-KO and Linsitinib-treated cells show comparable defects in IGF1 signaling (**new Figure 7c-f**) and on the *in vitro* oncogenic phenotypes (**Figure 7h-l**), support the conclusion that EHD1 depletion and IGF-1R inhibition impact the same signaling pathway. We have expanded our discussion to include these additional points.

Comment #2. The impact of reduced cell proliferation on tumorigenesis and malignancy should also be well discussed.

At first, I do not see any data mentioned for Lane 344-347. This data should be clearly indicated.

Furthermore, EHD1 KO and IGF-1 receptor inhibitors hardly activate IGF signaling. Wouldn't it be obvious that tumorigenesis would not be observed under the conditions which IGF signal is completely inhibited? For even normal cells it is difficult to proliferate or survive if IGF signal is blocked. I would like to see thorough discussion of this issue.

Response to Comment #2: We agree that loss of EHD1, similar to IGF-1R inhibition, strongly impacts the cell proliferation and survival and that these factors are likely to be key EHD1's role in tumorigenesis and metastasis. As the reviewer points out, our data indeed support the conclusion that upregulation of IGF-1R signaling is a major determinant of EHD1's pro-oncogenic function and we have further clarified this aspect in our **discussion(line #683-691)**. Our data supports the idea that all endpoints of IGF-1R signaling, not just proliferation, contribute to EHD1's pro-oncogenic role. This was a point we meant to convey from our migration/invasion assays that included a cell-proliferation inhibitor Mitomycin C (10 µg/ml) as described in the Methods section. To make this point clearer, we present the results of migration assays with and without Mitomycin C, which show that inhibition of cell proliferation is not a significant contributor to defective migration/invasion upon loss of EHD1(**new Supplementary Figure S2d**).

Comment #3. An EHD1-dependent mis-traffic of IGF-1R to the cell surface and sustaining it in the cell surface is a key element in the authors' scenario. But data illustrating the existence of IGF-1R in the cell surface seems to be relatively small in Figure 4B. I recommend a membrane fractionation experiment or surface biotinylation assay to strengthen your conclusion.

Response to Comment #3: Our conclusion that EHD1 regulates the cell surface IGF-1R traffic was based on analyses using FACS (**Figure 4f-h**) as well as confocal immunofluorescence (**Figure 5b-c, Supplementary Fig S7f, Fig S9, new Supplementary Fig. S10-11**) methods, where we observed a defect in the trafficking of the recycled as well as the newly-synthesized pools of IGF-1R to the cell surface. As suggested by the reviewer, we also performed a surface biotinylation assay (**new Figure 4j**) which shows a decrease in surface biotinylated IGF-1R in EHD1-KO TC71 and A673 cell lines and a rescue of this defect in KO cells with mEHD1 expression. Together, these results make a persuasive case for a requirement of EHD1 in IGF-1R traffic to the cell surface.

Comment #4. Fig 1D indicates that ~90% of clinical tissue samples had a high EHD1 expression, while only ~60% of them show high IGF-1R expressions (Fig 4G). Please provide an explanation regarding this gap.

Response to Comment #4: The reviewer raises a valid point that we had not addressed. While the exact reasons for this discrepancy are not known, we have added potential reasons for it in the **discussion** section (**line #764-774**). Our studies have identified additional receptor tyrosine kinase targets of EHD1 (CSF1R and EGFR) which may represent a subset of those RTKs regulated by EHD1. Studies of EWS have shown that while IGF-1R signaling is altered in a majority of cases³, aberrations of other RTKs are also found⁴. Thus, one plausible explanation for the discrepancy between EHD1 and IGF-1R co-overexpression could be that EHD1 regulates other RTKs in some EWS tumors. Another factor that might contribute to discrepancy between EHD1 and IGF-1R upregulation is that EWS-FLI-dependent aberrations in IGF-1R are multifactorial, including the upregulation of ligands or downregulation of inhibitory components⁵, and such factors may predominate in patients where IGF-1R levels themselves are not upregulated. Broader RTK analyses combined with EHD1 expression studies as well as concurrent analyses of multiple components of IGF-1R signaling network should help address these possibilities.

Comment #5. The authors mention that IGF-1R targeting clinical trials have been explored but the results have been disappointing. However, EHD1-KD/KO (presumably equivalent to IGF-1R inhibition) diminished tumor growth in transplanted mice, and pharmacological inhibition of IGF-1R indeed significantly decreased tumor malignancy (Fig 3 and 8). Please give the authors' opinion as to why IGF-1R targeting strategy has not been successful and therapeutic application of EHD1-dependent mechanism.

Response to Comment #5: The reviewer raises an important point. We have added a paragraph in the **discussion** (**line #775-791**) on our speculations on this key issue:

Given the EHD1 regulation of IGF-1R as well as other RTKs^{6, 7} as well as its targeting of additional pathways shown in other cancers^{8, 9, 10, 11, 12, 13, 14}, it appears likely that EHD1 targeting may still be viable and may potentially be combined with IGF-1R targeting. Additionally, the cell surface levels of RTKs not only dictate the levels of ligand-induced and kinase-dependent signaling as documented for IGF-1R in this study, but also dictate their kinase-independent signaling as demonstrated for several RTKs including EGFR and IGF-1R^{15, 16, 17, 18, 19, 20}. Such kinase-independent signaling has been linked to kinase inhibitor resistance²¹. Also, the expression of other RTKs provides a prevalent mechanism of resistance to RTK targeted therapies^{22, 23}, and the ability of EHD1 to target these could be an advantage. While detailed studies are needed to test this speculative model, the additive effects of linsitinib treatment and EHD1-KO in EWS cell lines on apoptosis or cell migration readouts (**Fig.7h-l**) are consistent with such an idea. Notably, in non-small cell lung cancer cell models, elevated levels of EHD1 correlated with insensitivity to EGFR inhibition and such insensitivity was overcome by genetic depletion of EHD1⁸.

Comment #6. In Figure 5B, how did the authors measure the ratio of IGF receptor recycling to the plasma membrane? Traffic to the recycling endosome is measured by co-localization with Rab11. The localization of the receptor to the plasma membrane should likewise be shown by co-localization with other co-staining.

And the immunostaining signal (Figure 5, 6, S5, S6) is too strong. So, what should appear as dots appear to be squashed. The signal should be weakened to clearly show the localization.

Response to Comment #6: In Figure 5B, the % IGF-1R transport to the cell surface was calculated as a ratio of the mean fluorescence intensity at the cell surface (using a freeform selection tool) to the mean fluorescence intensity of the entire cell. The background mean fluorescence was measured by selecting a region next to the cell of interest that showed no fluorescence and this value was subtracted from the cell fluorescence readings (included in Methods section). As suggested by the reviewer, we repeated these experiments with the inclusion of a plasma membrane marker- Alexa Fluor 647-conjugated Wheat Germ Agglutinin (WGA). The % colocalization of the pools of IGF-1R that colocalized with the cell surface WGA and intracellular RAB11 signals were assessed at 0, 30 min and 60 min time points post-endocytosis (of IGF-1R induced by IGF-1) by determining Pearson's correlation coefficient using ImageJ (**new Supplementary Figures S10-S11, staining controls in new Supplementary Figure S10e**). The immunostaining signals in the indicated figures have been dimmed down to improve their visualization as suggested by the reviewer.

Reviewer #2 Minor comments

4. Please provide more information of CCLE (Line#326).
5. Is what the sentence (Line#592-594) mention Fig 8? Is it correct?
6. Some abbreviations used in the manuscript appear without any explanations (e.g. ESFT, Line#103). Please read through again.

Response to Minor Comments: We have made the changes suggested by the reviewer.

Reviewer #3 Comments:

General/Major Comments of Reviewer #3. The paper by Chakraborty et al. investigates the role of EHD1 (EPS15 Homology Domain containing 1) protein in oncogenesis. The authors showed increased expression of EHD1 in patient samples, both primary tumors and metastasis. Use of shRNA and CRISPR mediated downregulation showed EHD1 role in proliferation, migration, invasion in vitro and tumor growth and metastasis in vivo. Mechanistically, the authors showed that EHD1 regulates surface localization of IGF1R (IGF1 receptor) and activation. The study is very well conducted and their results on co-targeting these two pathways is very translational. **There was no major concern** noted in the presented paper.

Response to Reviewer #3 Major Comments: We are thankful for the reviewer's enthusiasm for our findings and conclusions as well as their potential translational implications.

Reviewer #3 Minor concerns:

Comment #1. The authors should explain the shift in band of IGF-1R in Fig 4C. Is the phosphorylation mediated shift.

Response to Comment #1: This was a point also raised by Reviewer #1 (Comment #1). As indicated in our response to Reviewer #1 above, we have experimentally demonstrated the shift to be due to enhanced N-linked glycosylation of IGF-1R when EHD1 is overexpressed (**new Figure 4d-e, new Supplementary Fig. S7b**), and not due to changes in phosphorylation (**new Supplementary Figure S7c-d**).

Comment #2. The authors need to describe figure 8 in more detail (each panel). It is very hard to read.

Response to Comment #2: As suggested by the reviewer, we have expanded the description of the results presented in Fig. 8 to make it easier to comprehend (line 599-642).

REFERENCES

1. Kim JG, *et al.* Heterodimerization of glycosylated insulin-like growth factor-1 receptors and insulin receptors in cancer cells sensitive to anti-IGF1R antibody. *PLoS One* **7**, e33322 (2012).
2. Girnita L, *et al.* Inhibition of N-linked glycosylation down-regulates insulin-like growth factor-1 receptor at the cell surface and kills Ewing's sarcoma cells: therapeutic implications. *Anticancer Drug Des* **15**, 67-72 (2000).
3. Mora J, *et al.* Activated growth signaling pathway expression in Ewing sarcoma and clinical outcome. *Pediatr Blood Cancer* **58**, 532-538 (2012).
4. Jin W. The Role of Tyrosine Kinases as a Critical Prognostic Parameter and Its Targeted Therapies in Ewing Sarcoma. *Frontiers in Cell and Developmental Biology* **8**, (2020).

5. Prieur A, Tirode F, Cohen P, Delattre O. EWS/FLI-1 silencing and gene profiling of Ewing cells reveal downstream oncogenic pathways and a crucial role for repression of insulin-like growth factor binding protein 3. *Mol Cell Biol* **24**, 7275-7283 (2004).
6. Tom EC, *et al.* EHD1 and RUSC2 Control Basal Epidermal Growth Factor Receptor Cell Surface Expression and Recycling. *Mol Cell Biol* **40**, (2020).
7. Cypher LR, *et al.* CSF-1 receptor signalling is governed by pre-requisite EHD1 mediated receptor display on the macrophage cell surface. *Cell Signal* **28**, 1325-1335 (2016).
8. Huang J, *et al.* Targeting the IL-1 β /EHD1/TUBB3 axis overcomes resistance to EGFR-TKI in NSCLC. *Oncogene* **39**, 1739-1755 (2020).
9. Wang X, *et al.* NF- κ B-driven improvement of EHD1 contributes to erlotinib resistance in EGFR-mutant lung cancers. *Cell Death Dis* **9**, 418-418 (2018).
10. Liu Y, *et al.* Eps15 homology domain 1 promotes the evolution of papillary thyroid cancer by regulating endocytotic recycling of epidermal growth factor receptor. *Oncol Lett* **16**, 4263-4270 (2018).
11. Gao J, Meng Q, Zhao Y, Chen X, Cai L. EHD1 confers resistance to cisplatin in non-small cell lung cancer by regulating intracellular cisplatin concentrations. *BMC Cancer* **16**, 470 (2016).
12. Lu Y, Wang W, Tan S. EHD1 promotes the cancer stem cell (CSC)-like traits of glioma cells via interacting with CD44 and suppressing CD44 degradation. *Environ Toxicol* **37**, 2259-2268 (2022).
13. Liu Y, *et al.* A novel EHD1/CD44/Hippo/SP1 positive feedback loop potentiates stemness and metastasis in lung adenocarcinoma. *Clin Transl Med* **12**, e836 (2022).
14. Huang J, *et al.* A feedback circuit comprising EHD1 and 14-3-3 ζ sustains β -catenin/c-Myc-mediated aerobic glycolysis and proliferation in non-small cell lung cancer. *Cancer Lett* **520**, 12-25 (2021).
15. Cossu-Rocca P, *et al.* EGFR kinase-dependent and kinase-independent roles in clear cell renal cell carcinoma. *Am J Cancer Res* **6**, 71-83 (2016).
16. Thomas R, Weihua Z. Rethink of EGFR in Cancer With Its Kinase Independent Function on Board. *Front Oncol* **9**, 800 (2019).
17. Weihua Z, *et al.* Survival of cancer cells is maintained by EGFR independent of its kinase activity. *Cancer Cell* **13**, 385-393 (2008).
18. Tan X, Thapa N, Sun Y, Anderson RA. A kinase-independent role for EGF receptor in autophagy initiation. *Cell* **160**, 145-160 (2015).

19. Perrault R, Wright B, Storie B, Hatherell A, Zahradka P. Tyrosine kinase-independent activation of extracellular-regulated kinase (ERK) 1/2 by the insulin-like growth factor-1 receptor. *Cell Signal* **23**, 739-746 (2011).
20. Janku F, Huang HJ, Angelo LS, Kurzrock R. A kinase-independent biological activity for insulin growth factor-1 receptor (IGF-1R) : implications for inhibition of the IGF-1R signal. *Oncotarget* **4**, 463-473 (2013).
21. Ichim CV. Kinase-independent mechanisms of resistance of leukemia stem cells to tyrosine kinase inhibitors. *Stem Cells Transl Med* **3**, 405-415 (2014).
22. Garofalo C, *et al.* Efficacy of and resistance to anti-IGF-1R therapies in Ewing's sarcoma is dependent on insulin receptor signaling. *Oncogene* **30**, 2730-2740 (2011).
23. Remsing Rix LL, *et al.* IGF-binding proteins secreted by cancer-associated fibroblasts induce context-dependent drug sensitization of lung cancer cells. *Sci Signal* **15**, eabj5879 (2022).

REVIEWERS' COMMENTS:

Reviewer #1 (Remarks to the Author):

To authors,
The manuscript has been revised well. I think this manuscript will be acceptable.

Reviewer #2 (Remarks to the Author):

Authors have done a lot of experiments in a short time in response to reviewers comments, and authors have been able to respond to all the comments adequately. So reviewers consider that this paper is acceptable for publication in Communication Biology.

Reviewer #3 (Remarks to the Author):

accepted